# Kainate receptors regulate development of glutamatergic synaptic circuitry in the rodent amygdala

**Maria Ryazantseva[1,2], Jonas Englund[1,2], Alexandra Shintyapina[1,2], Johanna Huupponen[1,2], Vasilii Shteinikov[1], Asla Pitkänen[3], Juha M Partanen[1], Sari E Lauri[1,2]\***

[1]Molecular and Integrative Biosciences Research Program, University of Helsinki, Helsinki, Finland; [2]Neuroscience Center, University of Helsinki, Helsinki, Finland; [3]A. I. Virtanen Institute for Molecular Sciences, University of Eastern Finland, Kuopio, Finland

**Abstract** Perturbed information processing in the amygdala has been implicated in developmentally originating neuropsychiatric disorders. However, little is known on the mechanisms that guide formation and refinement of intrinsic connections between amygdaloid nuclei. We demonstrate that in rodents the glutamatergic connection from basolateral to central amygdala (BLA-CeA) develops rapidly during the first 10 postnatal days, before external inputs underlying amygdala-dependent behaviors emerge. During this restricted period of synaptic development, kainate-type of ionotropic glutamate receptors (KARs) are highly expressed in the BLA and tonically activated to regulate glutamate release via a G-protein-dependent mechanism. Genetic manipulation of this endogenous KAR activity locally in the newborn LA perturbed development of glutamatergic input to CeA, identifying KARs as a physiological mechanism regulating formation of the glutamatergic circuitry in the amygdala.

**\*For correspondence:**
sari.lauri@helsinki.fi

**Competing interests:** The authors declare that no competing interests exist.

## Introduction

Kainate-type of ionotropic glutamate receptors (KARs) are expressed in different cell types in various parts of the brain. While physiological roles for KARs in modulation of glutamatergic and GABAergic transmission have been described in the mature neural networks (*Lerma and Marques, 2013*), increasing evidence suggests predominant functions for KARs in the developing circuitry. KARs modulate formation of glutamatergic contacts by regulating neurite growth (*Ibarretxe et al., 2007*; *Joseph et al., 2011*; *Marques et al., 2013*), filopodial dynamics (*Chang and De Camilli, 2001*; *Tashiro et al., 2003*) and synaptic differentiation (*Marchal and Mulle, 2004*; *Lanore et al., 2012*; *Sakha et al., 2016*). At immature hippocampal synapses, KARs control efficacy and short-term dynamics of transmission by tonically regulating glutamate release probability (*Lauri et al., 2005*; *Lauri et al., 2006*). Interestingly, many of the developmental functions of KARs are mediated by non-canonical G-protein-dependent signaling mechanisms (*Lauri et al., 2006*; *Tashiro et al., 2003*; *Marques et al., 2013*). In the absence of the early KAR activity, development of the glutamatergic connectivity in the hippocampus is perturbed (*Vesikansa et al., 2012*; *Lanore et al., 2012*; *Orav et al., 2017*). Thus, the existing data indicate KARs as key modulators of synaptic transmission and plasticity during the critical window of synapse development in the hippocampus. However, the applicability of these mechanisms to other areas of the developing brain remains largely unknown.

Amygdala is part of the limbic system involved in memory, emotion and autonomic function. Basic architecture of the mammalian amygdala is present at birth, but similar to hippocampus, it undergoes structural and functional change across extended developmental period (*Tottenham and*

*Sheridan, 2009*). External inputs to basolateral amygdala (BLA) develop postnatally, the thalamic inputs being detectable around P7 (*Bouwmeester et al., 2002*), while cortical innervation emerges gradually between P10 and P21 (*Arruda-Carvalho et al., 2017*). Both lateral (LA) and basal amygdala (BA) nuclei have strong glutamatergic projections to the central amygdala (CeA), where the principal cells are GABAergic (*Sah et al., 2003*; *Pape and Pare, 2010*). However, practically nothing is known on the development of the intrinsic connections in the amygdala.

Here, we show that KAR subunits GluK1,2 and 5 are strongly expressed in the amygdala during early postnatal development, temporally coinciding with rapid development of functional glutamatergic synapses. In the newborn BLA, KARs are physiologically activated to regulate glutamate release probability via a G-protein-dependent mechanisms. Knockdown or overexpression of GluK1 expression locally in the newborn LA perturbed development of glutamatergic input to CeA, suggesting that endogenous KAR activity is critical for development of the LA-CeA connections. Interestingly, it was recently shown that enhanced activity of KARs in the adult amygdala, generated via GluK4 overexpression, leads to severe changes in the circuit excitability and amygdala dependent behaviors (*Arora et al., 2018*). Together, these data thus identify tonic KAR activity as a mechanism modulating behaviorally relevant synaptic circuitry in the amygdala.

## Results

### Kainate receptors are highly expressed in the BLA during the first postnatal week

Based on the existing literature, at least subunits GluK1, GluK2 and GluK5 are expressed in the adult rat BLA (*Bettler et al., 1990*; *Li et al., 2001*) where they have been implicated in regulation of synaptic transmission and plasticity (*Li et al., 2001*; *Ko et al., 2005*; *Shin et al., 2010*). To characterize the developmental profile for expression of various KAR subunits, BLA was dissected from acute rat brain slices at three developmental stages (postnatal day (P)4, P14, and P50). Absolute quantitative RT-PCR analysis of *Grik1-5* mRNAs, encoding for subunits GluK1-5, revealed that during early postnatal development (P4), the predominant KAR subunits in the BLA are *Grik1*, *Grik2* and *Grik5*, while *Grik3* and *Grik4* mRNAs were detected at low levels. During development, *Grik2* and *Grik5* mRNA expression remained relatively high (at P14 and P50, *Grik2*: $67 \pm 4\%$ and $69 \pm 10\%$ of the level at P4, respectively, ANOVA $F_{(2,21)}=4.0$, $p=0.034$; *Grik5*: $91 \pm 8\%$ and $163 \pm 15\%$, ANOVA $F_{(2,7)}=4.5$, $p=0.055$) while *Grik1* and *Grik4* mRNA levels were strongly downregulated already during the first two postnatal weeks (P14 and P50, *Grik1*: $32 \pm 5\%$ and $35 \pm 9\%$ of the level at P4, respectively, ANOVA $F_{(2,16)}=2.18$, $p<0.001$; *Grik4*: $54 \pm 6\%$ and $42 \pm 5\%$, ANOVA on ranks $H_{(2)}=19.39$, $p<0.001$). *Grik3* mRNA expression was undetectable at P14 and P50. Finally, the *Neto1* and *Neto2* mRNAs, encoding for KAR auxiliary subunits NETO1 and NETO2, were both detected in the neonatal BLA, *Neto2* representing the predominant subtype (*Figure 1A,B*).

To verify KAR expression at protein level, western blot was done from crude protein extracts from neonatal BLA using antibodies for GluK2/3 and GluK5. These antibodies have been previously validated using knockout tissue as a negative control (*Ruiz et al., 2005*; *Wyeth et al., 2014*). Unfortunately, selective specific antibodies for other KAR subunits are not available. Consistent with the qPCR data, GluK2/3 and GluK5 were clearly detected both at P4 and P14 BLA, at approximately the same levels (GluK2/3: ANOVA $F_{(1,4)}=5.48$, $p=0.079$; GluK5: ANOVA $F_{(1,4)}=2.72$, $p=0.174$; *Figure 1C*).

### KARs localize to both glutamatergic and GABAergic neurons in the neonatal amygdala

In situ hybridization (ISH) using fluorescent probes was used to study the pattern of KAR subunit expression in the amygdala in more detail. ISH signal in the various nuclei of the amygdala was analyzed in rat brain sections, cut from P4 and P14. In addition, sections from GAD67-GFP knock-in (*Gad1*$^{+/GFP}$) mice were analyzed to investigate the expression of various KAR subunits in GAD67GFP positive GABAergic neurons. In P4 *Gad1*$^{+/GFP}$ sections, GFP expression was detected in $10.8 \pm 0.6\%$, $13.7 \pm 0.9\%$ and $99 \pm 0.4\%$ of DAPI stained cells in LA, BA and CeA, respectively.

At P4, mRNAs encoding subunits GluK1, GluK2 and GluK5 were detected in all the amygdala nuclei analyzed (LA, BA and CeA) (*Figure 1D,E*). *Grik1* mRNA was observed in both GAD67GFP

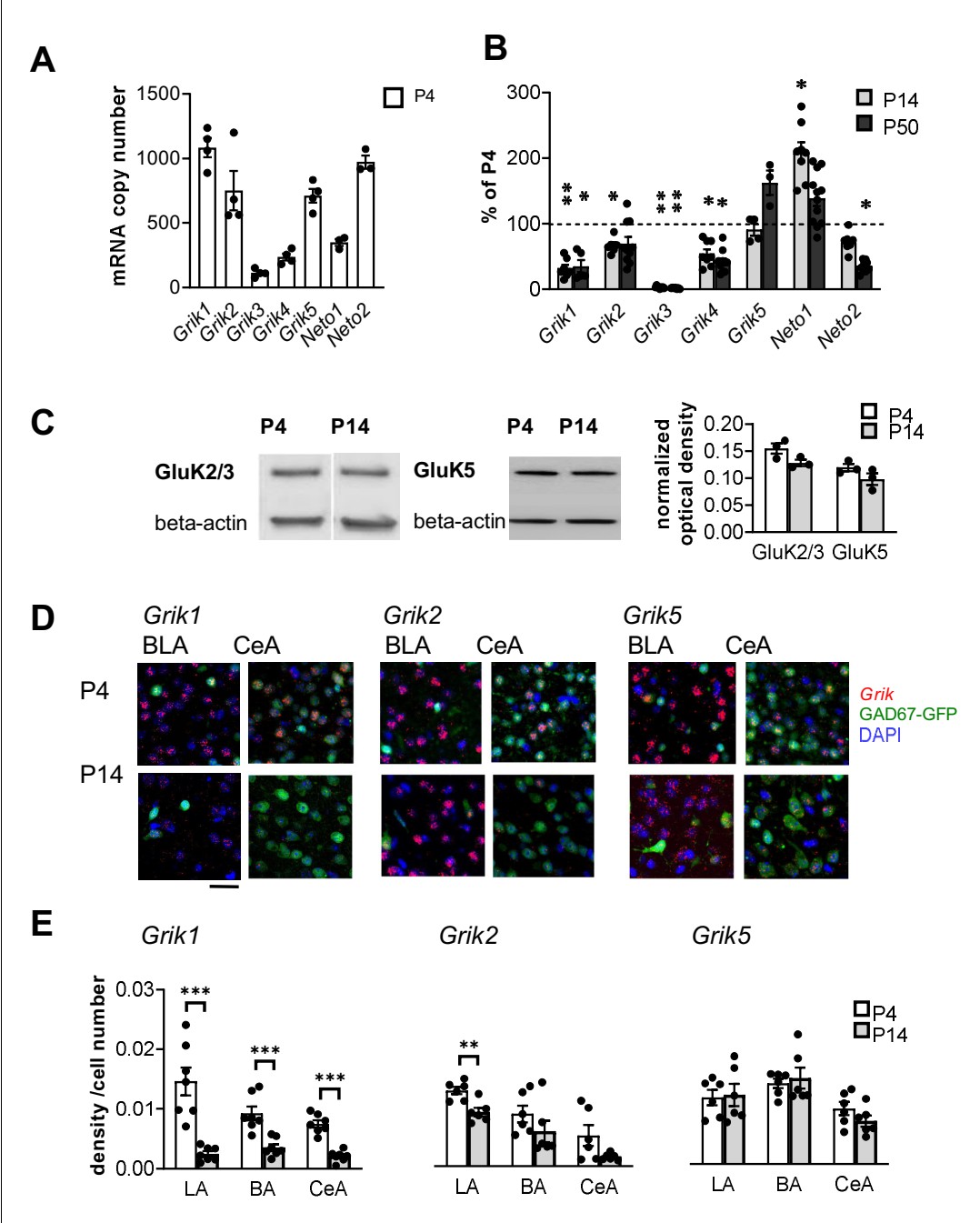

**Figure 1.** Kainate receptors are highly expressed in the amygdala during the first postnatal week. (**A**) The predominant KAR subunits expressed in the newborn BLA are GluK1, GluK2 and GluK5 and the auxiliary subunit NETO2. RT-aqPCR data on *Grik1-5, Neto1* and *Neto2* mRNA expression in the BLA at P4 (n = 3–4 rats/group). (**B**) RT-qPCR analysis of *Grik1-5, Neto1* and *Neto2* mRNA expression in the BLA at different stages of development, expressed as a percentage of the level at P4 (n = 3–12 rats/group). *Grik1* expression is strongly downregulated during early postnatal development. *p<0.05, **p<0.001 as compared to P4. (**C**) Western blot from neonatal (P4 and P14) BLA, using antibodies against GluK2/3 and GluK5. Beta-actin was used as a loading control. GluK2 and GluK5 protein was clearly detected in the BLA both at P4 and P14. Quantification of the signal indicates no significant developmental changes in the expression level. n = 3 rats/group. (**D**) Cell-type-specific expression pattern of KAR subunits in the amygdala during early postnatal development. Example images showing ISH staining (red) with antisense RNA probes against *Grik1, Grik2* and *Grik5* mRNA in the P4 mouse amygdala, in GAD67GFP positive (green) and negative (blue DAPI stain) cells. Scale bar, 100 µm. (**E**) Pooled data on the integrated density of the ISH signal, normalized to the number of cell bodies (identified by DAPI staining) within the analyzed region at P4 and P14. *Grik1* mRNA expression is strongly downregulated in all amygdala subnuclei during first weeks of postnatal development. LA, Lateral amygdala; BA Basal Amygdala; CeA Central amygdala (data for each subunit is pooled from n = 3–4 mice and three rats). *p<0.05, **p<0.001.

The online version of this article includes the following source data for figure 1:

**Source data 1.** Raw data for results shown in the *Figure 1*.

positive (GABAergic) and negative (likely representing glutamatergic principal neurons) cells in the BLA (25 ± 5% of GAD67GFP+ neurons at P4; *Figure 1D*). In the CeA, *Grik1* mRNA was detected in 30 ± 2% of the GAD67GFP-positive neurons. Consistent with the qPCR data, *Grik1* mRNA was significantly downregulated in the amygdala nuclei during the first postnatal weeks (ISH signal at P14 as percentage of P4, LA: 17 ± 3%, ANOVA on ranks, $H_{(1)}$=9.80, p<0.001; BA: 38 ± 6%, ANOVA $F_{(1,12)}$ = 19.93, p<0.001; CeA: 28 ± 4%, ANOVA $F_{(1,12)}$ = 60.87, p<0.001; *Figure 1E*).

*Grik2* mRNA was predominantly expressed in the GAD67GFP negative (glutamatergic) cells, and only detected in a minority of GABAergic neurons within the BLA (18 ± 3% of GAD67GFP+ neurons at P4; *Figure 1D*). In the CeA, *Grik2* mRNA was detected in 42% of GAD67GFP+ neurons (*Figure 1D*). *Grik2* mRNA expression was slightly downregulated during development, although this effect reached statistical significance only in the LA (ISH signal at P14 as percentage of P4, LA: 71 ± 10%, ANOVA, $F_{(1,10)}$=14.7, p=0.003; BA: 66 ± 37%, ANOVA on ranks, $H_{(1)}$=2.08, p=0.18; CeA: 31 ± 8%, ANOVA on ranks, $H_{(1)}$=2.08, p=0.18; *Figure 1E*).

*Grik5* mRNA was detected in both GAD67GFP positive and negative cells (30 ± 2% of GAD67GFP+ neurons at P4) in the BLA. *Grik5* mRNA was also strongly expressed in 43 ± 6% GAD67GFP+ neurons in the CeA at P4 (*Figure 1D*). At P14, the same overall pattern of staining remained, except for slight but not significant downregulation of *Grik5* expression level in the CeA (ISH signal at P14 as percentage of P4, LA: 103 ± 15%, BA: 107 ± 12%; CeA: 77 ± 9%, ANOVA $F_{(1,10)}$=2.06, p=0.18; *Figure 1E*).

## Early postnatal development of glutamatergic synaptic connectivity in the amygdala

The high developmental expression of KAR subunits in the amygdala is similar to that in the hippocampus, where KARs have specific developmentally restricted functions related to maturation of the glutamatergic synaptic connectivity. However, the exact time course for development of glutamatergic synaptic inputs to various nuclei in the amygdala is not known. Therefore, we recorded miniature excitatory postsynaptic currents (mEPSCs) from LA, BA and CeA neurons to get a general view on the functional glutamatergic inputs at different stages of postnatal development. The recorded cells were filled with biocytin for post-hoc analysis of the spine density.

Interestingly, the mean mEPSC frequency, reflecting the density of functional glutamatergic inputs to the recorded neuron, increased with a distinct developmental time course at different nuclei. In the LA and BA, the mEPSC frequency increased significantly from P4-7 to P9-11 (LA: 479 ± 88%, ANOVA on ranks, $H_{(1)}$=10.42, p=0.001; BA: 493 ± 95%, ANOVA on ranks, $H_{(1)}$=10.38, p=0.001), and from P9-11 to P14-16 (LA: 240 ± 50%, ANOVA $F_{(1,15)}$=6.69, p=0.021; BA: 196 ± 35%, ANOVA $F_{(1,16)}$=4.60, p=0.048), but reached a plateau at the second postnatal week (*Figure 2A,B*). In contrast, in the CeA the mEPSC frequency increased rapidly between P4-7 and P9-11 (353 ± 62%, ANOVA on ranks, $H_{(1)}$=7.78, p=0.005), after which no significant changes in the mean frequency of the events were detected within the analyzed time window (*Figure 2A,B*). The mEPSC amplitudes remained stable throughout the early postnatal development (not shown).

Spine maturation takes place later than formation of a functional synapse. Visualization of dendritic spines of the biocytin filled neurons revealed that the mean spine density stabilized earlier in the CeA as compared to BLA (spine density, % increase P14-16 vs P20-21, LA: 149 ± 13%, ANOVA $F_{(1,14)}$=5.30, p=0.037; BA: 166 ± 25%, ANOVA $F_{(1,12)}$=5.19, p=0.042; CeA: 94 ± 7%, ANOVA $F_{(1,14)}$=0.149, p=0.71; *Figure 2C,D*). This is consistent with the idea that the glutamatergic synapses to CeA form rapidly during the first two postnatal weeks in rats, while the inputs to BLA develop with an extended time course.

## Ionotropic KAR function is not detected in the BLA during the first postnatal week

The developmental peak in KAR, and particularly GluK1 expression was associated with the time of rapid development of glutamatergic synapses in the BLA and CeA. To understand the physiological roles of KARs in the developing circuit, we mapped functional ionotropic KARs in the neonatal amygdala using agonist application during voltage clamp recordings. Surprisingly, application of kainate (KA, 2 µM) in the presence of selective antagonists for AMPA, NMDA and GABA receptors (2 µM NBQX, 50 µM APV and 100 µM PiTX, respectively) induced no or only a very small current (LA: 6 ± 1

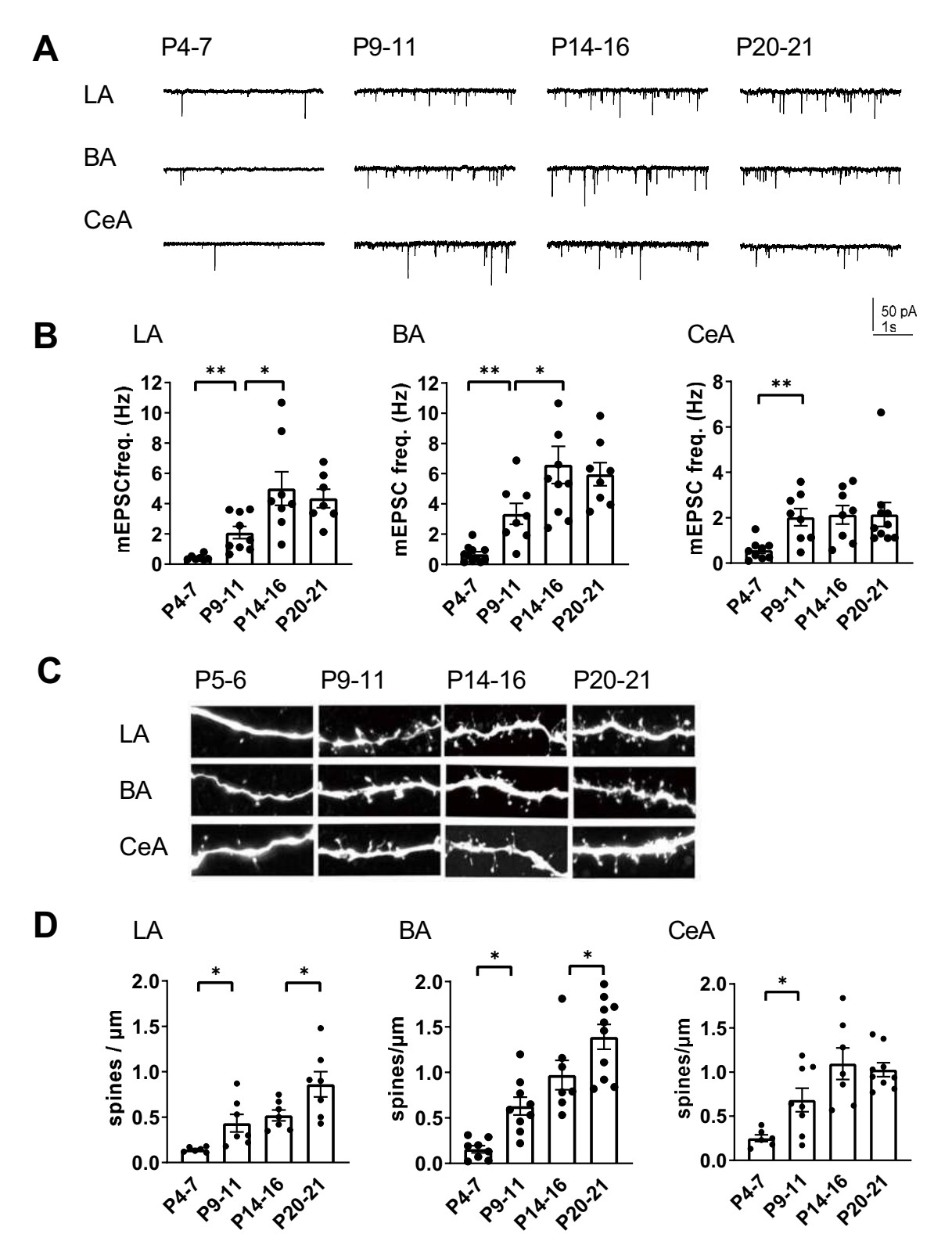

**Figure 2.** Development of glutamatergic synaptic connectivity in the amygdala. (**A**) Example traces of action-potential independent glutamatergic events (mEPSCs), recorded from LA, BA and CeA neurons at different stages of development. (**B**) Pooled data of mEPSC frequency in amygdala subnuclei at different stages of development (n = 7–11 from 4 to 9 rats/group). The mEPSC frequency increases with a different developmental time course in CeA as compared to BA and LA. *p<0.05, **p<0.001. (**C**) Confocal images illustrating dendritic spines in biocytin filled neurons from LA, BA

*Figure 2 continued on next page*

*Figure 2 continued*

and CeA at P5, P10, P15 and P21 (D) Pooled data on spine density in LA, BA and CeA neurons at different stages of development. Data is obtained from 6 to 10 dendrites from three rats/group. *p<0.05; **p<0.001.

The online version of this article includes the following source data for figure 2:

**Source data 1.** Raw data for results shown in the *Figure 2.*

pA; BA: 8 ± 1 pA) in the BLA neurons during the first postnatal week (P4-P7), while a marked inward current was observed later on in development (P14-P16) (LA: 81 ± 19 pA, BA 37 ± 5 pA) (*Figure 3A, B*). In the CeA, a small KA-induced inward current was detected already at P4-P6 (15 ± 2 pA), but the current was significantly larger at P14-16 (27 ± 3 pA, ANOVA on ranks, $H_{(1)}$=8.00, p=0.005) (*Figure 3A,B*). Since kainate might cause rapid desensitization of the receptors, we also used another KAR agonist, domoate to confirm the result in the first postnatal week (P4-P7). Application of domoate (500 nM) induced no current in LA neurons (4 ± 1 pA), but in BA a small inward current was detected (17 ± 1 pA; *Figure 3C,D*). Thus, the high expression of the KAR subunits in the newborn LA does not correlate with ionotropic function of KARs, which emerges after the second postnatal week. Small KAR-dependent currents were detected in BA and in particular, in the CeA already at P4-P6, but in contrast to the declining mRNA expression profile, the ionotropic function increased towards the second postnatal week.

## KARs regulate glutamatergic transmission to BA and CeA during the first postnatal week

At immature CA3-CA1 synapses, presynaptic GluK1 subunit containing KARs tonically regulate glutamate release probability in a G-protein and PKC-dependent manner (*Lauri et al., 2006*; *Sallert et al., 2007*). To test whether KARs might have similar functions in the newborn amygdala, we studied the effect of 200 nM ACET, a selective antagonist for GluK1 subunit containing KARs (*Dargan et al., 2009*), on mEPSCs in LA, BA and CeA during first (P4-7) and second postnatal weeks (P10-16). In LA, ACET had no significant effect on mEPSCs at P4-6 (frequency 92 ± 14% of control, paired t-test $t_{(6)}$=-0.80, p=0.45, amplitude 101 ± 8% of control) or at P14-P16 (frequency 91 ± 3%, paired t-test $t_{(5)}$=2.56, p=0.051; amplitude 103 ± 4% of control; *Figure 4A*). In contrast, ACET application resulted in a small but significant increase in the mEPSC frequency in BA at P4-P6 (130 ± 4% of control, signed rank test, Z = 2.93, p=0.004), but not at P14-P16 (92 ± 7%, paired t-test, $t_{(3)}$=1.11, p=0.35) (*Figure 4B*). ACET had no effect on mEPSC amplitude at either developmental time point (97 ± 1% and 100 ± 4% of control, respectively). The developmentally restricted effect of GluK1 KAR antagonism on mEPSC frequency in BA is similar to that previously characterized in the area CA1 of hippocampus (*Lauri et al., 2006*), indicative of tonic KAR-mediated inhibition of glutamate release.

In CeA, ACET application resulted in a marked depression of mEPSC frequency in first postnatal week (P5-6) (73 ± 5%, paired t-test, $t_{(10)}$=4.22, p=0.0018) without affecting their amplitude (99 ± 3%). Also this effect was developmentally restricted, as ACET had no effect of mEPSCs at P10-P11 CeA (frequency 101 ± 7%, amplitude 98 ± 4%; *Figure 4C*).

These data show that GluK1 subunit containing KARs are physiologically activated to regulate glutamatergic transmission in the amygdala during the first postnatal week. Their effect is area and/or cell type specific, with KAR-mediated inhibition and facilitation of transmission at glutamatergic inputs to BA and CeA, respectively.

## Transmission at immature BLA – CeA synapses is tonically facilitated by presynaptic G-protein-coupled KARs

The data on KAR expression and function at the neonatal amygdala so far show 1) high expression of GluK1, GluK2 and GluK5 in the BLA 2) low ionotropic KAR activity in the BLA and 3) KAR-dependent regulation of mEPSC frequency in the BA and CeA. One interpretation of these data is that GluK1 subunit containing KARs, expressed in the principal cells of LA (and BA), are targeted to axons to regulate transmitter release at emerging projections to BA and CeA via a mechanism that does not require ionotropic activity.

To investigate this, we focused to study the pathway from BLA to CeA at P5-6, and recoded EPSCs from CeA in response to electrical stimulation of the LA. Consistent with hypothesis that

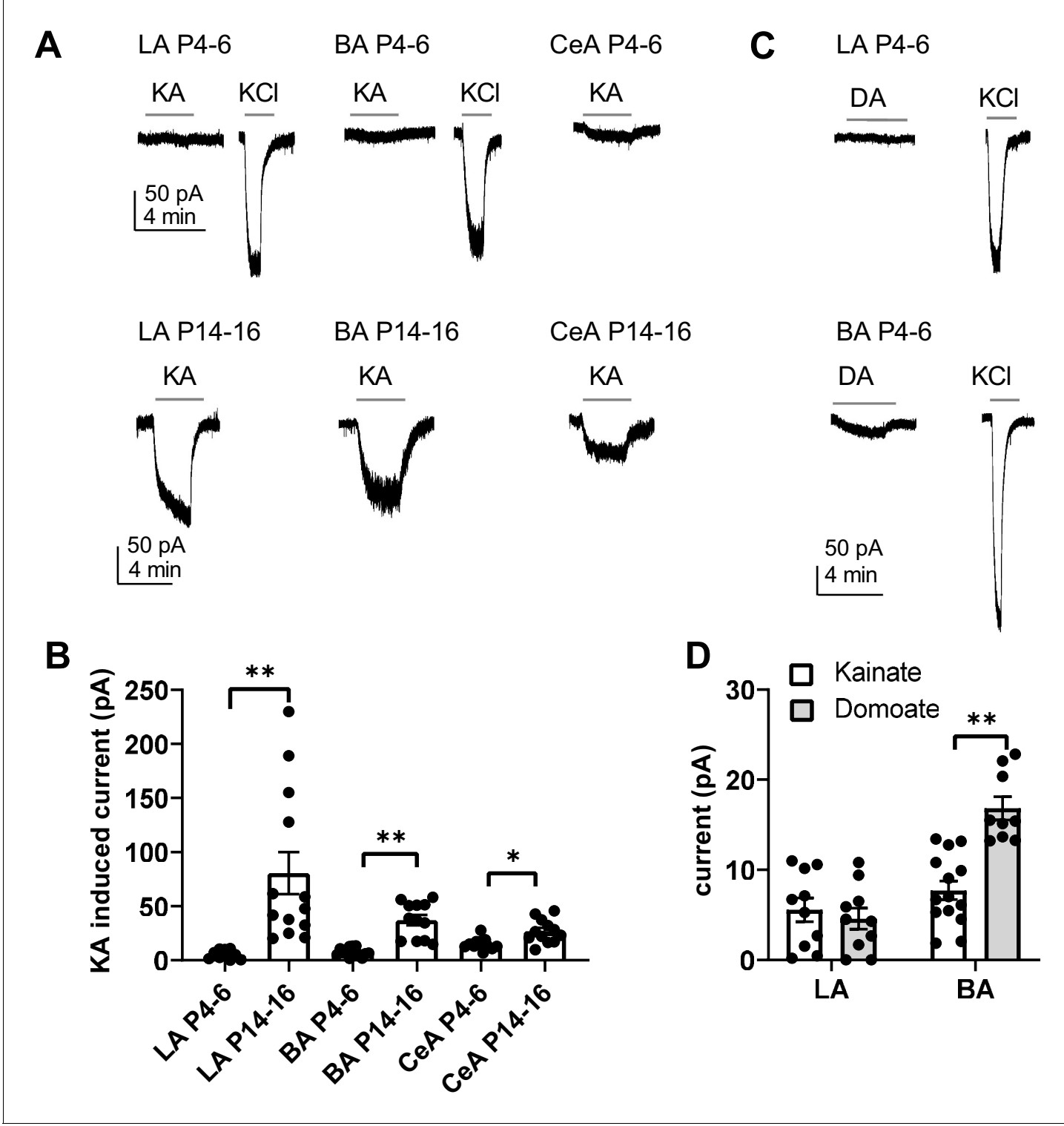

**Figure 3.** Functional map of ionotropic KAR activity in the amygdala during the first postnatal week. (A) Example traces illustrating inward currents in response to application of 2 µM kainate (KA) in the presence of blockers for AMPA, NMDA and GABA-A receptors, in LA, BA and CeA neurons at P4-6 and P14-16. 33 mM KCl was applied as a positive control. In the neonatal BLA, no KA-induced currents are detected while a clear current is induced under similar conditions at P14-P16. (B) Pooled data on the effect of 2 µM kainate on holding current in LA, BA and CeA neurons at two different stages of development (P4-6 and P14-16; LA n = 10 [6] and 13[7]; BA: n = 14 [7] and 12 [6], CeA n = 12[10] and 12[8], respectively). (C) Example traces showing that application of 500 nM of domoate (DA) induced no current in neonate LA neurons, while a small inward current was detected in BA (P4-6). The recordings were made in the presence of blockers for AMPA, NMDA and GABA-A receptors. (D) Pooled data comparing the effect of 2 µM kainate (LA

*Figure 3 continued on next page*

*Figure 3 continued*

n = 10[4], BA n = 14[4]) and 500 nM domoate (LA: n = 10[4]; BA: n = 9[4]) on holding current in neonatal LA and BA neurons. The n numbers in this and following figure legends refer to the number of experiments, followed by the number of animals in brackets.

The online version of this article includes the following source data for figure 3:

**Source data 1.** Raw data for results shown in the *Figure 3*.

presynaptic GluK1 subunit containing KARs regulate transmission at this pathway, ACET application resulted in significant reversible depression of the EPSC amplitude (62 ± 5% of control, paired t-test, $t_{(7)=}$-3.586, p=0.009) that was associated with an increase in paired – pulse ratio (PPR) (150 ± 14%, paired t-test, $t_{(8)}$=4.01, p=0.0039) (*Figure 5A*). Furthermore, EPSCs evoked by single pulse or paired pulses with 50 ms interval of BLA stimulation were completely blocked by the AMPA-receptor selective antagonist GYKI53655 (25 µM) (8 ± 2% of control, paired t-test, $t_{(7)=}$-3.679, p=0.008), excluding the possibility that postsynaptic KARs significantly contributed to the eEPSC in this pathway (*Figure 5B*). Adult CeA neurons also receive glutamatergic input from extra-amygdalar brain regions, including sensory input from the auditory cortex and thalamus (*Keifer et al., 2015*). However, during the first postnatal week, we could not observe any postsynaptic responses in CeA neurons to stimulation of external and internal capsulae, the regions containing cortical and thalamic projections to amygdala (not shown). This suggests that external inputs to CeA are not present during the first postnatal week and thus do not contribute to the observed changes in mEPSC frequency in response to KAR antagonism in the neonate (P5-6) CeA.

Pharmacological tools were used to further characterize the mechanism involved. Application of a broad KAR agonist KA (500 nM) and ATPA (1 µM), agonist selective for GluK1 subunit containing KARs, resulted in transient increase in mEPSC frequency in P5-P6 CeA (KA: 146 ± 27%, paired t-test, $t_{(6)}$=4.24, p=0.0055; ATPA: 191 ± 21%, paired t-test $t_{(6)}$=3,14, p=0.02; *Figure 5C,D*), with no effect on mEPSC amplitude (91 ± 8.8% for KA and 100 ± 13% for ATPA). The peak increase in mEPSC frequency in response to ATPA and KA was comparable in size or if anything, slightly larger in ATPA as compared to KA (ANOVA, $F_{(1,12)}$ = 4.50, p=0.055), suggesting that GluK1 subunit containing receptors can fully account for KAR regulation of transmission at this synapse without additional (GluK1 lacking) KAR populations involved.

Finally, ACET had no effect on mEPSCs in P4-P6 CeA in slices that were preincubated in presence of pertussis toxin (PTX), a selective inhibitor of the $G\alpha_{i/o}$ signaling (mEPSC frequency in ACET, 108 ± 17% of control; paired t-test $t_{(4)}$=1.243, p=0.282; *Figure 5E*). The incubation procedure itself had no effect on the regulation of transmission by KARs, as ACET application resulted in significant depression of mEPSC frequency in slices incubated under control conditions (72 ± 12%, paired t-test, $t_{(4)}$=2.934, p=0.043). Together, these data indicate that GluK1-subunit containing KARs upregulate glutamate release at immature BLA-CeA synapses via a G-protein coupled signaling mechanism.

## Loss of GluK1 in the amygdala in a cKO mouse model impairs glutamatergic innervation and maturation of CeLA neurons

Immature-type KAR activity is proposed to be critical for development and fine-tuning of the glutamatergic synapses (*Vesikansa et al., 2007*; *Vesikansa et al., 2012*; *Orav et al., 2017*). To investigate the significance of the early KAR activity on the development and maturation of glutamatergic transmission in the amygdala, we focused on studying the LA-CeA connections. To this end, we generated a conditional (floxed) mouse model for *Grik1* (*Grik1* cKO), which allowed us to inactivate GluK1 expression selectively in the neonate BLA in vivo with local injection of GFP-CRE encoding AAV virus at P2-P4. The viral transduction covered 79 ± 5% and 61 ± 8% of cells in LA and BA, respectively, and 10 ± 4% of cells in the CeA, based on green fluorescence signal in DAPI labeled cells at P21 (*Figure 6A*).

The consequences of targeted GluK1 knockout on glutamatergic synapses were studied with patch clamp recordings of mEPSCs from acute slices at P21, followed by *post hoc* morphological analysis of the recorded biocytin filled neurons. The recordings were made from GFP-CRE expressing (GluK1 KO) and GFP expressing (control) neurons in the LA as well as from their target neurons in the centrolateral (CeLA) part of the CeA, which, in contrast to <P16, can be clearly visualized at this developmental stage in acute slices.

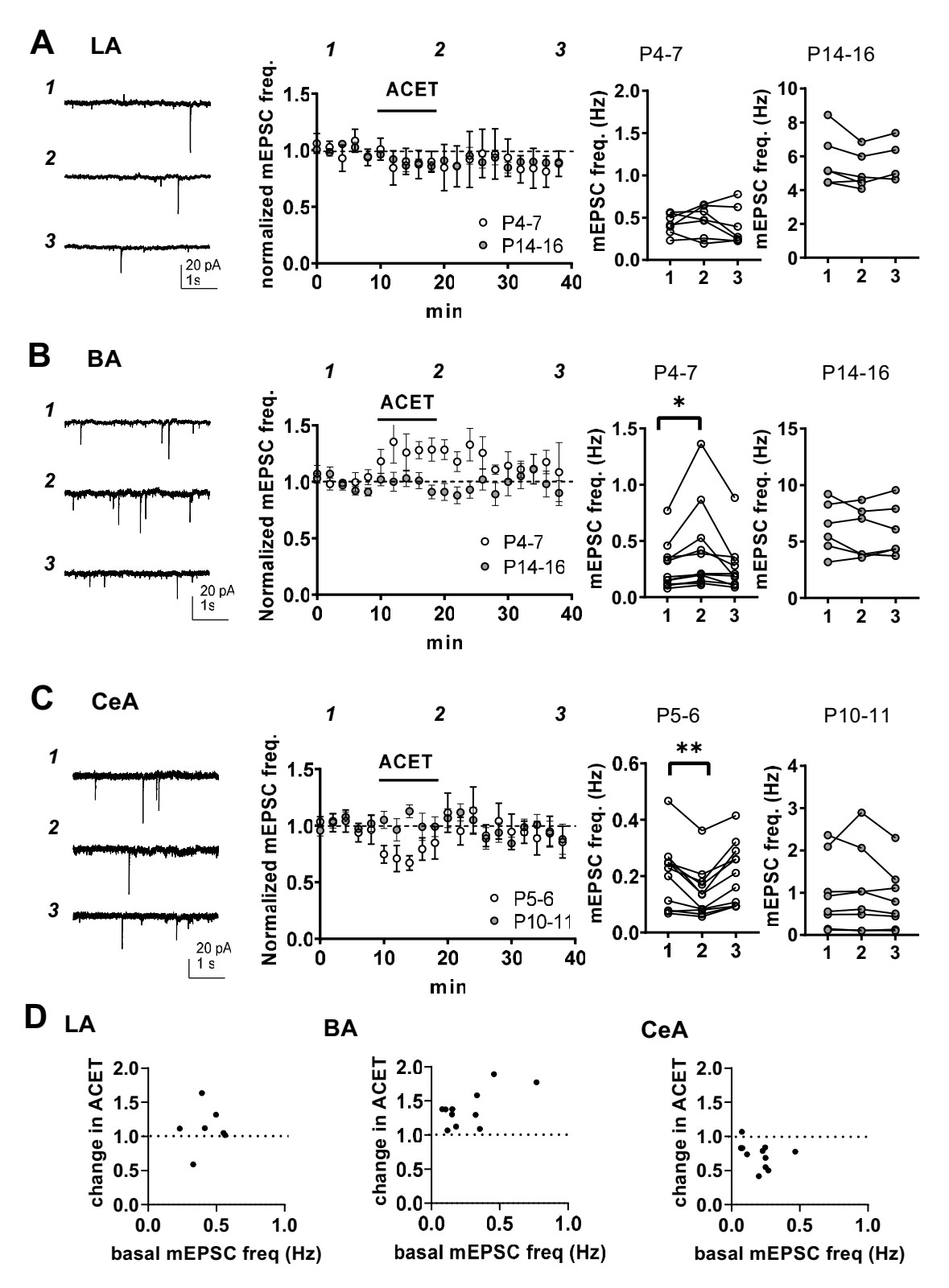

**Figure 4.** KARs regulate glutamatergic inputs to BA and CeA during the first postnatal week. (**A**) KAR antagonism does not affect action-potential independent glutamatergic events (mEPSCs) to LA during the first postnatal week. Example traces (P5) from the time points indicated (1-3) and time course plot of averaged data (± s.e.m.) illustrating little or no effect of ACET (200 nM) on mEPSC frequency in LA at P4-7 and at P14-16. The graphs on the right show the mEPSC frequency under control conditions (1), in the presence of ACET (2), and after wash-out of the drug (3), in individual

*Figure 4 continued on next page*

*Figure 4 continued*

experiments at two different stages of development (P4-7 and P14-16; n = 7 [4] and 6 [5], respectively). (**B**) Similar data for recordings in the BA. In newborn BA, ACET application reversibly increases mEPSC frequency (n = 11[8]). This effect is no longer observed at P14-P16 (n = 6[4]). *p<0.05. (**C**) Similar data for recordings in the CeA. In the first week of life (P4-6; n = 9[6]) but not at P9-11 (n = 6[4]), ACET application leads to robust depression of mEPSC frequency. **p<0.01. (**D**) Scatter plots illustrating the effect of ACET on mEPSC frequency in individual cells in various amygdala subnuclei in the first postnatal week. The P4-7 data from A-C is plotted as a function of the basal mEPSC frequency.

The online version of this article includes the following source data for figure 4:

**Source data 1.** Raw data for results shown in the *Figure 4*.

Loss of GluK1 had little or no effect on glutamatergic inputs to LA neurons, since there was no significant difference in the mEPSC frequency or amplitude between GFP-CRE and GFP- expressing neurons (frequency 100 ± 9% of GFP control, ANOVA $F_{(1,23)}$=0.0004, p=0.99; amplitude 96 ± 3% of GFP control, ANOVA $F_{(1,23)}$=0.21, p=0.65; *Figure 6B*). In contrast, in CeLA neurons, both mEPSC frequency and amplitude were significantly lower in GFP-CRE slices as compared to controls (frequency 38 ± 6% of GFP control, ANOVA on ranks, $H_{(1)}$=7.752, p=0.005; amplitude 73 ± 6% of GFP control, ANOVA $F_{(1,19)}$ = 10.226, p=0.005; *Figure 6B*). *Post hoc* morphological analysis indicated that the loss of mEPSCs in the CeLA neurons was associated with a significant reduction in the spine density (61 ± 12% of GFP control, ANOVA $F_{(1,22)}$ = 10.19, p=0.004; *Figure 6C*) and interestingly, lower dendritic length (54 ± 7% of GFP control, ANOVA $F_{(1,19)}$ = 10.41, p=0.004; *Figure 6D*) and branching (two-way ANOVA, $F_{(19,418)}$=2,89, p<0,0001; *Figure 6D*). No differences in dendritic morphology were observed between GFP-CRE and GFP expressing neurons in the LA (*Figure 6—figure supplement 2*).

These data pinpoint GluK1 in maturation of CeLA but not LA neurons. Following GluK1 knockout, the spine density and dendritic morphology in CeLA neurons was comparable to those at P10 in wildtypes (*Figure 6—figure supplement 3*), suggesting that absence of GluK1 disrupted developmental maturation of the dendritic tree in CeLA neurons. Consistent with this idea, inactivation of GluK1 expression in BLA at a later developmental stage (BLA virus injection at P14, ex vivo recording at P35) had no effect on mEPSCs (mEPSC frequency: 111 ± 10% of control, ANOVA $F_{(1,15)}$=0.411, p=0.531; amplitude: 106 ± 7% of control, ANOVA on ranks, $H_{(1)}$=0.454, p=0.50; *Figure 6E*), spine density (90 ± 8% of control, ANOVA $F_{(1,17)}$=0.452, p=0.51; *Figure 6F*), dendritic length (107 ± 8% of control, ANOVA $F_{(1,16)}$=0.283, p=0.602; *Figure 6G*) or dendritic intersections (two-way ANOVA $F_{(15,195)}$=0.7235, p=0.76, *Figure 6G*) in CeLA neurons.

## Loss of GluK1 in the amygdala perturbs glutamatergic synaptic transmission and network excitability at the LA- CeLA circuitry

To understand the consequences of targeted GluK1 knockout on glutamatergic projections from LA to CeLA in more detail, we studied the properties of synaptic transmission using evoked responses. Specifically, we investigated the input/output relationship, AMPA/NMDA ratio and paired pulse ratio (PPR) of EPSCs, recorded from CeLA in response to LA stimulation in slices from *Grik1* cKO mice (P21) that were injected with GFP and GFP-CRE encoding AAV viruses in BLA at P2-4. The stimulus-response relationship of EPSCs, evoked by LA stimulation using fixed intensities, was steeper in control (GFP) slices as compared GFP-CRE group (two-way ANOVA, $F_{(16, 48)}$=4.54, p<0.0001; *Figure 7A*). In addition, the AMPA/NMDA ratio was significantly lower in the GFP-CRE slices as compared to controls (39 ± 8% of control, ANOVA on ranks, $H_{(1)}$=4.973, p=0.026; *Figure 7B*). There was no difference in the paired pulse ratio (105 ± 18% of control, ANOVA $F_{(1,18)}$ = 0.0572, p=0.814; *Figure 7C*). Thus, loss of GluK1 in the BLA is associated with significantly weaker AMPA-R-mediated synaptic transmission from LA to CeLA.

Finally, the significance of targeted GluK1 knockout on amygdala excitability was assessed by recording spontaneous EPSCs and IPSCs from CeLA. Regular firing (RF) and late firing (LF) neuron populations were analyzed separately given the recent data suggesting that overexpression of KARs has dissimilar effects on glutamatergic inputs to these neurons (*Arora et al., 2018*). We found that developmental loss of GluK1 specifically affected glutamatergic inputs to the LF cell population, where sEPSC frequency was significantly lower in the GFP-CRE slices as compared to GFP controls (sEPSC frequency, 56 ± 5%, ANOVA $F_{(1,22)}$=5.54, p=0.028; *Figure 7D*). There was no significant difference in the sEPSCs between the groups in the RF cells (GFP-CRE, 95 ± 14% of control, ANOVA

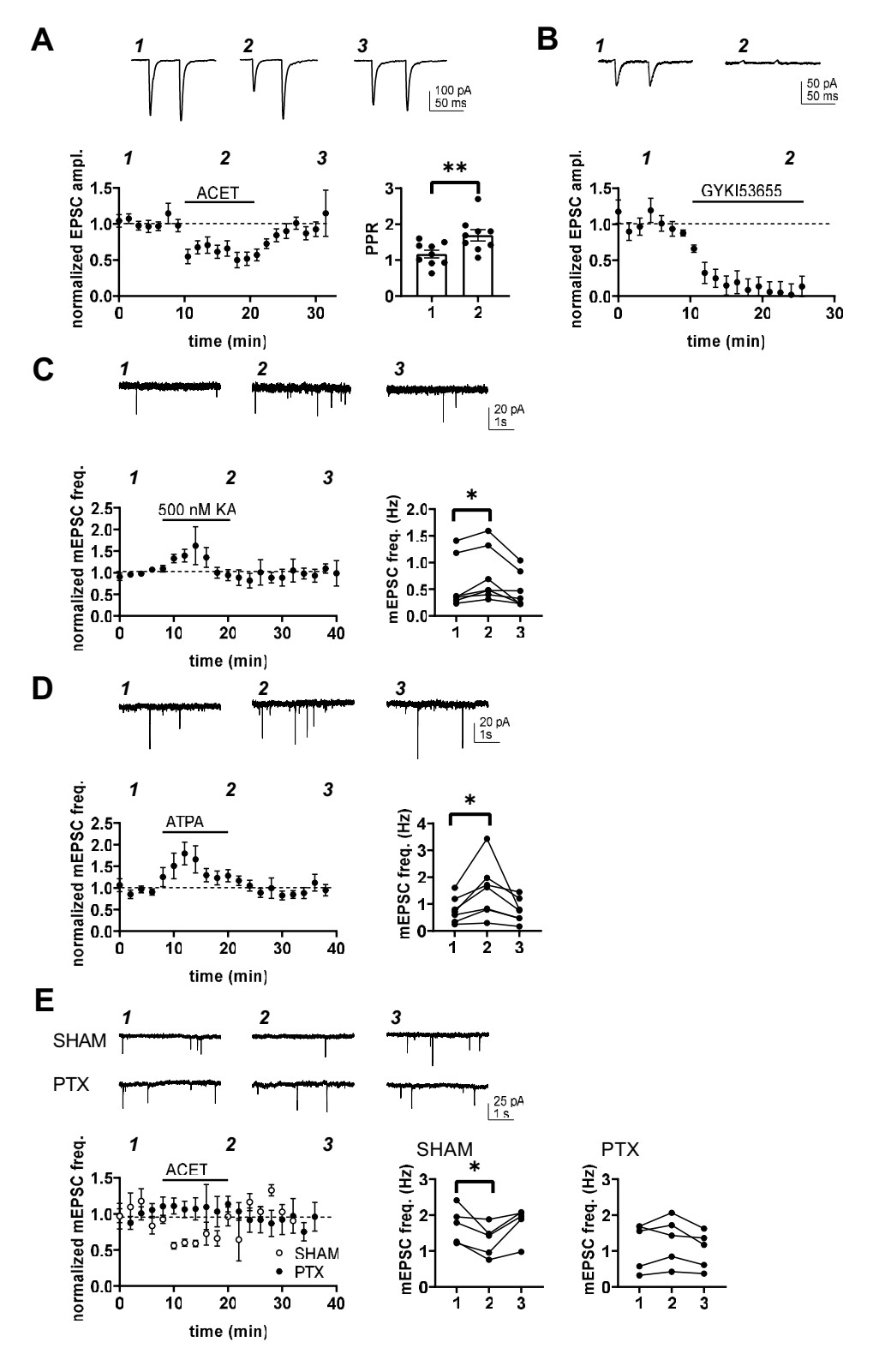

**Figure 5.** Transmission at immature BLA-CeA synapses is tonically facilitated by presynaptic G-protein-coupled KARs. (A) KAR antagonism leads to reversible depression of glutamatergic transmission from to BLA-CeA during first postnatal week. Example traces of EPSCs, recorded from CeA neurons at P5 in response to paired-pulse stimulation of the LA, under control conditions and in the presence of 200 nM ACET. Pooled data (± s.e.m.) showing that ACET application is associated with a reduction in the EPSC amplitude and an increase in paired pulse ratio. n = 9[5]; **p<0.001. (B)

*Figure 5 continued on next page*

*Figure 5 continued*

Postsynaptic KARs do not contribute to transmission at immature BLA-CeA synapses. Similar data as in A, showing that the AMPA receptor antagonist GYKI53655 fully blocks the EPSC. n = 8 (4); p<0.001. (C) Example traces from the time points indicated (1-3) and time course plot (average ± s.e.m.) depicting a transient increase mEPSC frequency in response to application of KA (500 nM). The graphs show mEPSC frequency under control conditions (1), in the presence of KA (2), and after wash-out of the drug (3) in individual cells in CeA (P4-6). n = 7 (6); *p<0.05. (D) Application of the GluK1 selective agonist ATPA leads to an increase in mEPSCs frequency, similar to that observed in response to 500 nM KA. Example traces and pooled data as in C. n = 7 (4); *p<0.05. (E) The tonic KAR activity regulating glutamatergic transmission to CeA acts via a G-protein-dependent mechanism. The effect of ACET on mEPSC frequency is fully blocked in slices that are pre-incubated in the presence pertussis toxin (PTX; n = 5[3]), but not in SHAM incubated controls (n = 5[3]). *p<0.05.

The online version of this article includes the following source data for figure 5:

**Source data 1.** Raw data for results shown in the *Figure 5*.

on ranks, $H_{(1)}$=1.76, p=0.184). sIPSC frequency was slightly but not significantly higher in RF neurons in GFP-CRE slices as compared to controls (LF: 115 ± 25%; ANOVA on ranks, $H_{(1)}$=0.97, p=0.33; RF: 173 ± 32%, ANOVA on ranks, $H_{(1)}$=3.21, p=0.073; *Figure 7D*).

These data indicate that loss of GluK1 weakens glutamatergic projections from LA to CeLA, leading to significant changes in the excitation-inhibition balance in the amygdala circuit. Specifically, GluK1 deficiency was associated with a significantly lower E/I ratio in the late firing neurons in the CeLA (sEPSC/sIPSC ratio in GFP-CRE and GFP groups, LF neurons: 9.9 ± 2 and 4.4 ± 0.6, ANOVA on ranks, $H_{(1)}$=4.91, p=0.027; RF neurons: 6.3 ± 2.4 and 4.8 ± 1.7, ANOVA on ranks, $H_{(1)}$=0.48, p=0.488).

## Presynaptic GluK1 KARs guide development of glutamatergic synaptic input to the CeA

To confirm that the observed effects at CeLA neurons were indeed due to the early endogenous activity of GluK1 in the LA, we used lentiviral vectors to manipulate GluK1 expression in neonatal rats. Lentiviral shRNA and overexpression produce faster changes in gene expression as compared to AAV – CRE-mediated inactivation of a floxed allele, allowing the consequences of P4-5 injection to be analysed already at P10-11. In addition, the injected lentiviruses produced a transduction that was spatially restricted to LA (GFP expression as a percentage of all DAPI-positive cells, LA: 59 ± 5%, BA: 0.7 ± 0.4%, CeA: 0.1 ± 0.4%; *Figure 8A*), thus selectively targeting the presynaptic/axonal GluK1 receptors in the LA-CeA projections.

In slices with local lentiviral expression of *Grik1* shRNA in the LA, mEPSC frequency in CeA at P10-P11 was significantly lower as compared to controls expressing a mock shRNA construct (38 ± 18%, ANOVA, $F_{(1,9)}$=7.59, p=0.02; *Figure 8B*). Accordingly, GluK1 overexpression (oe) in the LA was associated with significantly higher mEPSC frequency in the CeA as compared to GFP expressing controls at P14 (188 ± 38%, ANOVA on ranks, $H_{(1)}$=4.77, p=0.029; *Figure 8C*). Neither, *Grik1* shRNA or GluK1 oe had a significant effect on mEPSC amplitude (71 ± 12%, ANOVA $F_{(1,9)}$=3.31, p=0.106; and 91 ± 11%, ANOVA $F_{(1,19)}$ = 0.45, p=0.51, respectively). Importantly, GluK1 oe in the LA recapitulated the tonic KAR-dependent regulation of mEPSC frequency in the CeA at a developmental stage (P14) when this regulation was no longer observed at control or GFP expressing slices. Thus, in slices with LA-specific GluK1 oe, ACET application resulted in a significant decrease in the mEPSC frequency in CeA cells (73 ± 11% of control, paired t-test, $t_{(4)}$=5,496, p=0.0053), but had no effect in control (GFP expressing) slices (103 ± 7% of control, paired t-test $t_{(4)}$=1,267, p=0.274) (*Figure 8D*). These results confirm that presynaptic GluK1 KARs, located in the axons of LA principal neurons, regulate glutamatergic transmission to CeA.

The ongoing GluK1-dependent facilitation of transmission contributes to the observed increase in the basal mEPSC frequency in GluK1 oe slices. In addition, morphological analysis of the CeA neurons indicated that the number of dendritic intersections in both apical and distal dendrites was significantly higher in GluK1 oe slices as compared to controls (total dendritic length in GluK1 oe slices, 200 ± 22% of GFP, ANOVA $F_{(1,11)}$=22.6, p<0.001; dendritic intersections, two-way ANOVA $F_{(8, 90)}$=3.014, p=0.0049; *Figure 8E*). In contrast, no significant differences in the density of dendritic spines were detected (mean density of spines in GluK1 oe slices, 116 ± 7% of GFP control; ANOVA $F_{(1,14)}$=1.48, p=0.24; *Figure 8F*).

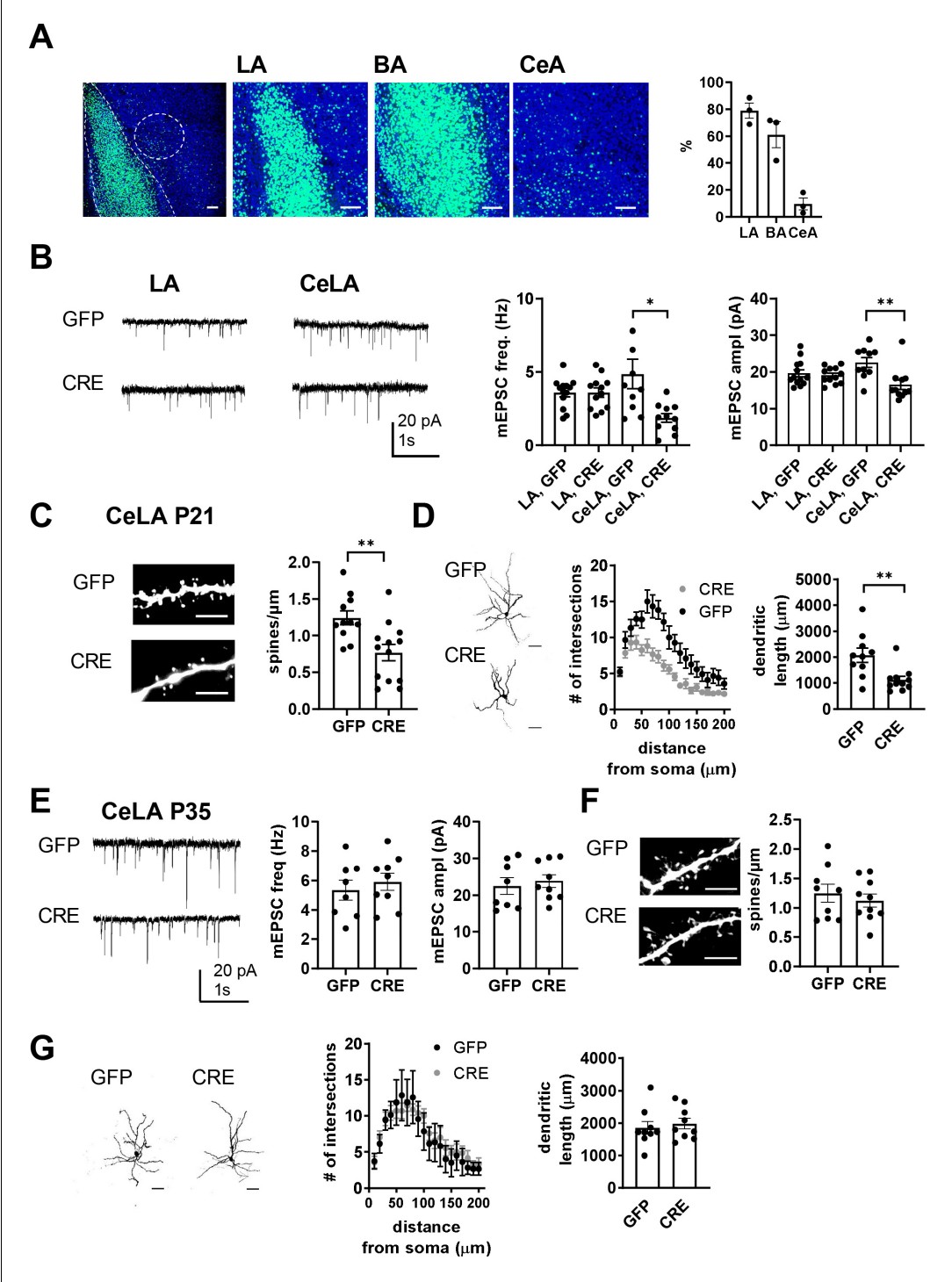

**Figure 6.** Loss of GluK1 in the amygdala in a cKO mouse model impairs glutamatergic innervation and maturation of CeLA neurons. (**A**) Image illustrating the GFP fluorescence in the BLA and CeA of *Grik1* cKO mice (P21), after in vivo injection of GFP-CRE AAV virus at P2. The viral transduction mainly targeted LA and BA (79 ± 5% and 61 ± 8% of the DAPI labeled cell bodies (blue), respectively), but also some cells in the CeA (10 ± 4% of cells). Scale bar, 100 μm. (**B**) Example traces and pooled data showing that loss of GluK1 in the BLA has no effect on the mEPSCs at LA neurons but results in lower mEPSC frequency and amplitude in the CeLA as compared to controls. AAV viruses encoding for GFP-CRE or GFP (control) were injected to the BLA at P2-4, and mEPSCs were recorded from acute slices at P21. LA: GFP n = 13(5), GFP-CRE n = 12(4); CeLA: GFP n = 10(3), GFP-CRE n = 11(3); *p<0.05. (**C**) *Post hoc* morphological characterization of the biocytin filled CeLA neurons (P21). Density of dendritic spines was significantly lower in GFP-CRE slices as compared to controls (GFP-CRE, n = 11(4); GFP n = 13(4)). **p<0.01. Scale bar, 5 μm (**D**) Sholl analysis of the same neurons. The number of dendritic intersections and the total dendritic length is significantly lower in GFP-CRE slices as compared to controls. Pooled data (GFP-CRE,

*Figure 6 continued on next page*

Figure 6 continued

n = 11(4); GFP n = 10(4)) and confocal images (z-stack) illustrating the morphology of the dendritic tree. **p<0.01. Scale bar, 50 µm **E)** Local inactivation of *Grik1* at a later developmental stage has no effect on mEPSCs and dendritic morphology of CeLA neurons. AAV viruses encoding for GFP-CRE or GFP (control) were injected to the BLA of the *Grik1* cKO mice at P14 and ex vivo analysis was done at P35. The image panels show mEPSCs (**E**), dendritic spines (**F**), intersections and total dendritic length (**G**) for 7-9(3) neurons from both GFP and GFP-CRE slices.

The online version of this article includes the following source data and figure supplement(s) for figure 6:

**Source data 1.** Raw data for the results shown in the *Figure 6*.
**Figure supplement 1.** Validation of the GluK1 cKO mouse model.
**Figure supplement 2.** Morphological characterization of GFP and GFP-CRE expressing LA neurons in *Grik1* cKO mice.
**Figure supplement 3.** Development of CeA neuron morphology in control mice and in cKO mice lacking GluK1 in the BLA.

Together, these data indicate that developmental manipulation of GluK1 expression in the LA results in lasting changes in glutamatergic synaptic transmission from LA to CeA and support that presynaptic GluK1 subunit containing KARs are required for appropriate development of intrinsic glutamatergic connections in the amygdala.

## Discussion

Given the importance of the amygdala in several developmentally originating neuropsychiatric disorders, it is surprising how little is understood on the mechanisms that govern formation and refinement of the synaptic circuitry in the amygdala. In particular, very little is known on the development of the projections from the BLA to CeA, critical elements in the neural circuits mediating fear and anxiety (*Duvarci and Pare, 2014*; *Herry and Johansen, 2014*). Here, we show that functional glutamatergic connections to CeA develop rapidly during the first 10–11 postnatal days in rats, before the external cortical inputs to amygdala emerge (*Bouwmeester et al., 2002*; *Arruda-Carvalho et al., 2017*). Further, we demonstrate that during the time of intense synaptogenesis, transmission to CeA is strongly regulated by physiological activity of presynaptic KARs, which is critical for appropriate development of the LA-CeA synapses.

Histological and functional evidence indicate that maturation of the intrinsic and extrinsic connectivity in the rodent BLA occurs postnatally, being completed around P21-P28 (*Nair and Gonzalez-Lima, 1999*; *Moryś et al., 1999*; *Bosch and Ehrlich, 2015*; *Ryan et al., 2016*; *Arruda-Carvalho et al., 2017*), in parallel with emergence of amygdala – dependent forms of associative emotional learning (e.g. *Landers and Sullivan, 2012*; *Hartley and Lee, 2015*; *Deal et al., 2016*). Consistently, we found that spine density at BLA neurons increased gradually during the first 3 postnatal weeks while the density of functional inputs, assessed by mEPSC frequency, stabilized after the second postnatal week. Interestingly, glutamatergic inputs to the CeA reached maturity earlier as compared to BLA, the mEPSC frequency stabilizing already at P10-11 and spine density at around P14-16. Based on tracing studies, the external inputs to amygdala develop gradually from P7 onwards, most cortical inputs arriving only after the second postnatal week (*Bouwmeester et al., 2002*; *Arruda-Carvalho et al., 2017*). However, CeA receives strong glutamatergic input from BLA (*Pape and Pare, 2010*; *Sah et al., 2003*), suggesting that rapid increase in mEPSC frequency between P5-P10 reflects mainly formation of the intrinsic amygdala circuits and specifically, development of the BLA – CeA connectivity. These data support that the local circuit in the amygdala is wired before the cortical inputs underlying behavior actuate.

Kainate receptors are highly expressed in the adult amygdala (*Bettler et al., 1990*; *Li et al., 2001*). However, no previous data on the expression and function of KARs during amygdala development exists. Our data show that the low-affinity subunits GluK1 and GluK2 and the high-affinity subunit GluK5 are strongly expressed in the BLA during the first postnatal week, when the glutamatergic synapses are rapidly developing. Early in development, all the subunits were also detected in the central amygdala. The expression pattern of various KAR subunits in the newborn amygdala showed no evident cell-type specificity, thus giving little insight into their functions in the neonatal BLA circuitry. Yet, the finding that GluK1 expression was strongly downregulated during development, in parallel with maturation of synaptic connectivity, is reminiscent to hippocampus where this subunit has central role in regulating synaptic plasticity and formation of CA3-CA1 synaptic connections (*Lauri et al., 2006*; *Vesikansa et al., 2012*; *Clarke et al., 2014*; *Orav et al., 2017*).

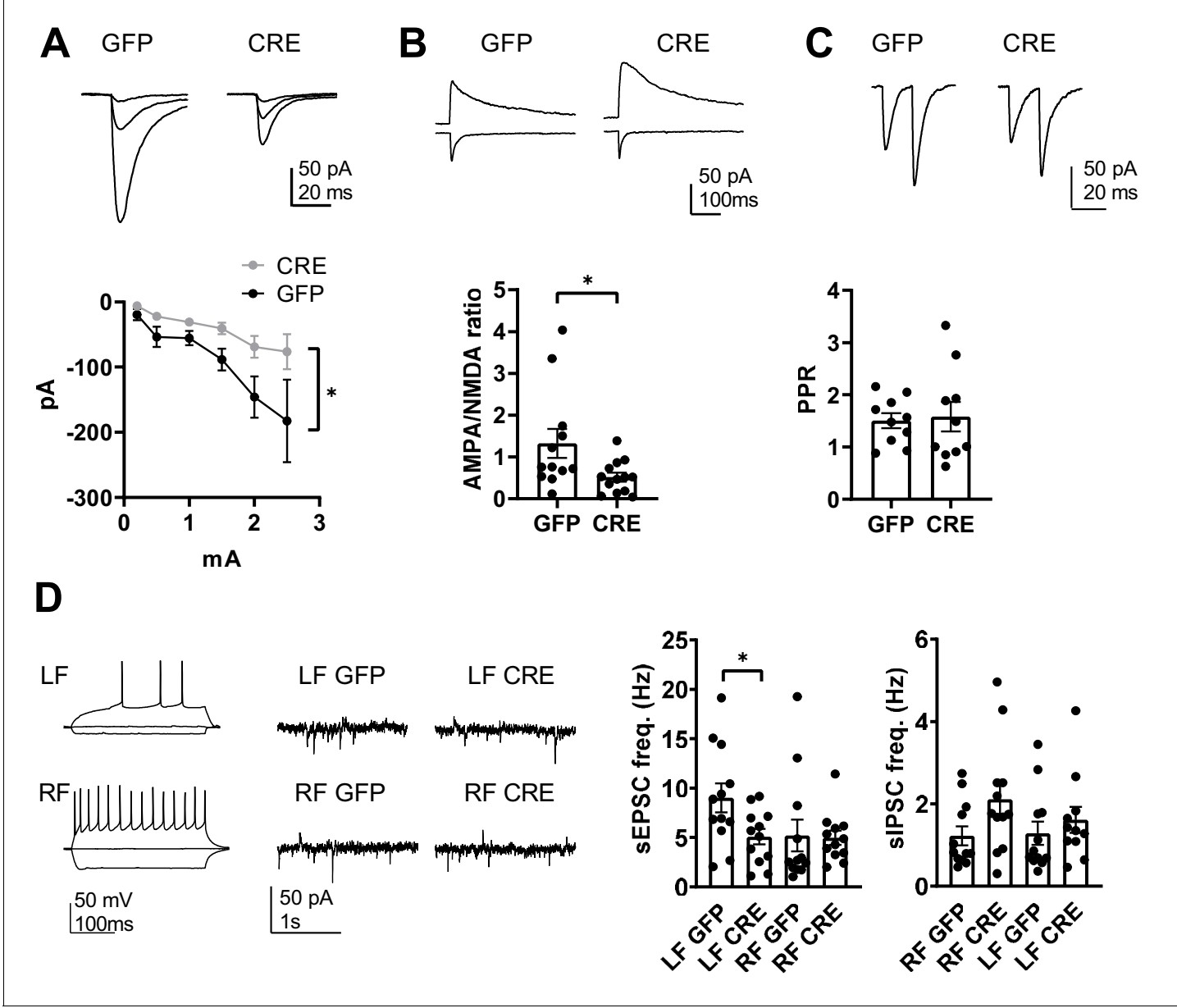

**Figure 7.** GluK1 deficiency in the amygdala perturbs glutamatergic transmission at LA-CeLA and associates with lower E/I ratio at CeLA late-firing neurons. (A) Input – output characteristics of EPSCs, recorded from CeLA neurons in response to LA stimulation. Recordings for all the data presented in panels A-D were made in acute slices from P21 *Grik1* cKO mice, injected with GFP or GFP-CRE encoding AAV viruses in BLA at P2-4. The EPSCs were significantly smaller in slices where GluK1 was locally inactivated as compared to controls. Example traces for responses evoked by three different stimulation intensities (0.2, 1 and 2.5 mA) for both groups and pooled data (GFP, n = 9[5], GFP-CRE n = 8[5], *p<0.05). (B) Example traces and pooled data illustrating the amplitude ratio of AMPAR and NMDAR–mediated EPSCs, evoked by LA stimulation in CeLA neurons. AMPA/NMDA ratio was significantly smaller in GFP-CRE group as compared to GFP controls (GFP n = 12[5], GFP-CRE n = 13[5], *p<0.05). (C) Paired –pulse ratio of LA-CeLA EPSCs was not different between GFP-CRE and GFP slices. Example traces and averaged data on paired-pulse ratio (PPR) of EPSCs (GFP, n = 10[4], GFP-CRE n = 10[4]). (D) Example traces of firing pattern of late-firing (LF) and regular-firing (RF) neurons in CeLA (P21) in response to depolarizing current pulses and recordings of spontaneous IPSCs and EPSCs under conditions where the sIPSCs are seen as outward currents and the sEPSCs are directed inwardly. sEPSC frequency was lower in LF neurons in CeLA in animals infected with GFP-CRE encoding AAV viruses at P2-4 as compared to GFP controls. Pooled data on frequencies of sEPSCs and sIPSCs in RF and LF neurons in CeLA (n = 12[4] for both groups, *p<0.05).

The online version of this article includes the following source data for figure 7:

**Source data 1.** Raw data for the results shown in the *Figure 7*.

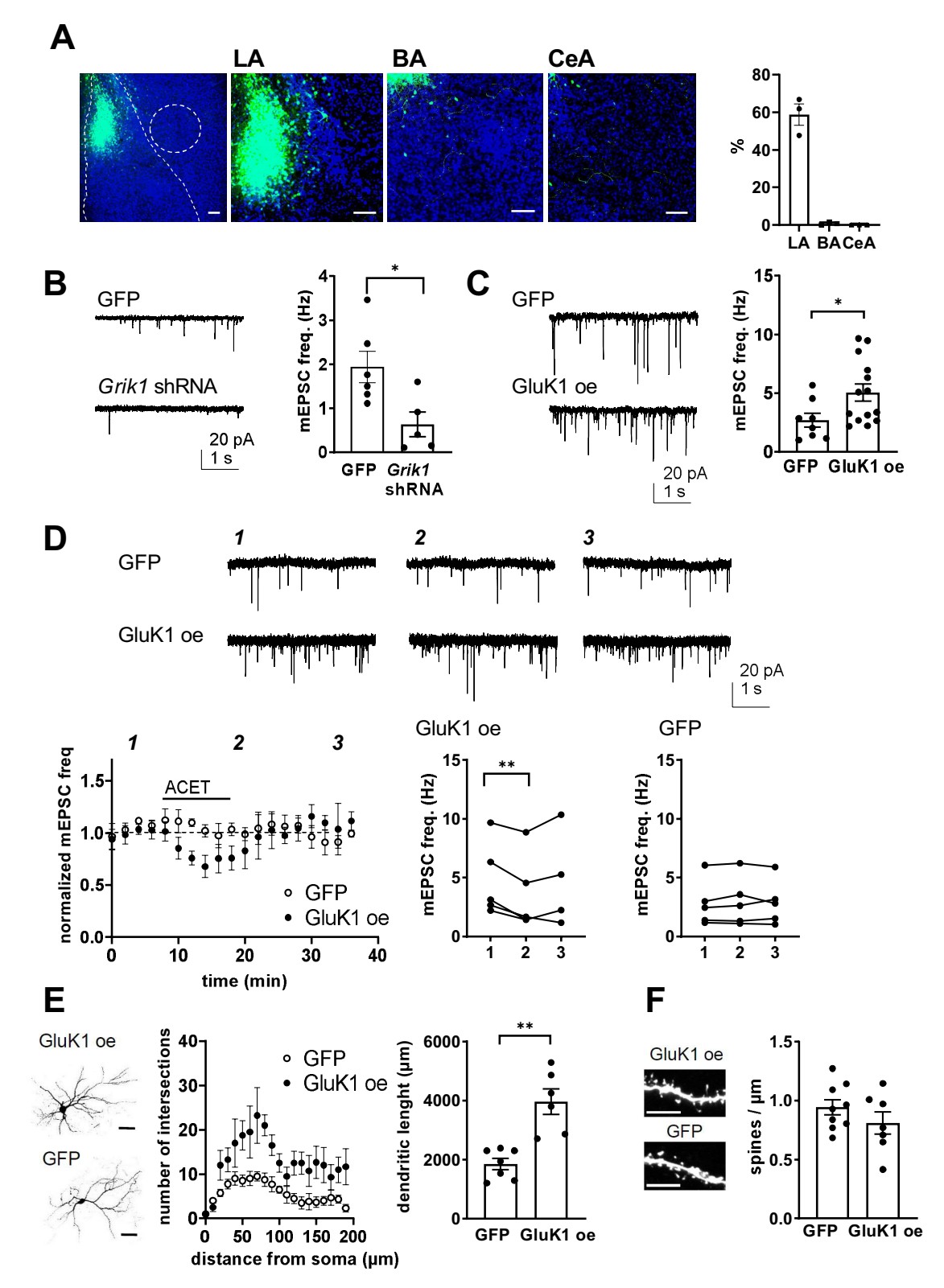

**Figure 8.** Presynaptic GluK1 KARs guide development of glutamatergic synaptic input to CeA. (**A**) Image showing GFP expression specifically in the LA at P11, after in vivo injection of *Grik1* shRNA/GFP encoding lentivirus at P5. The inset panel shows a magnified image illustrating GFP expression on average of 59 ± 5% of DAPI labeled (blue) cells in the LA, while practically no expression was detected in the BA (0.7 ± 0.4%) and CeA (0.1 ± 0.4%). Scale bar, 100 μm. (**B**) Example traces and pooled data showing that shRNA-mediated GluK1 knockdown in the LA results in lower mEPSC frequency in

*Figure 8 continued on next page*

*Figure 8 continued*

the CeA as compared to controls. *Grik1* shRNA or mock lentivirus was injected to LA at P4-5 and the recordings were made from CeA neurons at P10-11 (GFP; n = 6 (5); *Grik1* shRNA/GFP: n = 5 (3) *p<0.05). (C) Slices with GluK1 overexpression (GluK1 oe) in the LA show a significantly higher mEPSC frequency in the CeA as compared to controls (GFP). In vivo transduction was done at P6-9 and recordings were made at P14-15. GluK1 oe, n = 14(5), GFP n = 8 (4); *p<0.05. (D) GluK1 oe in the LA recapitulates the developmental phenotype of presynaptic KAR activity. Example traces from the time points indicated (1-3) and time course plots of pooled data illustrate that in GluK1 oe slices, ACET application causes a significant decrease in mEPSC frequency in the CeA neurons (n = 5 (3), p<0.05). At this developmental stage (P14-15), ACET has no effect on mEPSCs in control (GFP infected) slices (n = 5 (3)). Graphs on the right show mEPSC frequency under control conditions (1), in the presence of KA (2), and after wash-out of the drug (3) in individual cells. *p<0.05, paired t-test. (E) GluK1 oe results in an increase in dendritic length and branching. Sholl analysis of the biocytin filled CeA neurons indicate that number of dendritic intersections and the total dendritic length is significantly increased in slices with GluK1 oe in the LA. Pooled data (GluK1 oe, n = 6 (3); GFP n = 7 (3)) and confocal images (z-stack) illustrating the morphology of the dendritic tree. **p<0.01. Scale bar, 50 μm. (F) GluK1 oe does not influence the density of dendritic spines in CeA neurons. Pooled data of the mean density of dendritic spines (GluK1 oe n = 7 (4); GFP, n = 9 (4)). Example images shown on the right. Scale bar, 5 μm.

The online version of this article includes the following source data and figure supplement(s) for figure 8:

**Source data 1.** Raw data for the results shown in the *Figure 8*.
**Figure supplement 1.** Validation of the *Grik1* shRNA.

Intriguingly, during the first postnatal week corresponding to the developmental peak in expression of most KAR subunits, ionotropic activity of KARs was weak in the BLA. Contrary to the declining mRNA expression profile, KAR-mediated inward currents were significantly increased during development in all amygdala nuclei studied. Instead, we found that tonic G-protein-coupled signaling via GluK1 subunit containing KARs regulated glutamate release in a developmentally restricted manner. Our data thus supports that G-protein-coupled 'non-canonical' signaling is a predominant mode of KAR activity during the time of intense synaptogenesis, while the emergence of ionotropic KARs correlates with circuit maturation (*Marchal and Mulle, 2004*).

In adult BLA, KAR activity facilitates glutamate release (e.g. *Aroniadou-Anderjaska et al., 2012*; *Arora et al., 2018*) indicating that presynaptic KARs are present in adult but are functionally distinct from the immature-type GluK1 subunit containing receptors that tonically inhibit glutamate release in the BA. In the CeA, tonic KAR-dependent facilitation of glutamate release was detected during the first postnatal week of life. According to our results, presynaptic KARs are not endogenously active in the juvenile (P14-16) amygdala; however, previous studies using KAR antagonists with a broader subunit specificity (UBP302, UBP310) have detected endogenous KAR activity also in the adult BLA (*Aroniadou-Anderjaska et al., 2012*; *Arora et al., 2018*). During first weeks of life, concentration of ambient glutamate is high (e.g. *Hanson et al., 2019*) and sufficient to activate high-affinity KARs (*Lauri et al., 2006*; *Segerstråle et al., 2010*). The developmental increase in glial glutamate uptake thus provides a plausible mechanism for the decline in the tonic KAR activity. Parallel changes in KAR subunit composition affecting subcellular localization and affinity of the receptors (e.g. *Vesikansa et al., 2012*) likely contribute to the switch from immature to adult type KAR activity. The previous findings showing coupling of GluK1 to G-proteins (*Rutkowska-Wlodarczyk et al., 2015*) and recapitulation of immature-type synaptic signaling by GluK1c overexpression (*Vesikansa et al., 2012*; present data) support the view that expression of GluK1 and in particular, the GluK1c splice variant, is central to the early metabotropic KAR activity. In addition, the high agonist affinity, permitting the tonic activation, depends on availability of subunits GluK4/5 as well as the auxiliary subunits NETO1/2 reviewed by *Lerma and Marques (2013)*.

Recently, it was demonstrated that overexpression of KAR subunit GluK4 enhances tonic KAR activity in the adult amygdala and leads to profound changes in the circuit excitability (*Arora et al., 2018*). Since the immature-type tonic KAR activity temporally coincided with rapid development of functional glutamatergic synapses in the amygdala, we presumed that the physiological role of this activity is to modulate development of the amygdala circuitry. Consistently, local GluK1 knockdown in the LA during the time of rapid development of the BLA-CeA connectivity, but not later on in life, significantly impaired glutamatergic transmission from LA to CeA. Loss of GluK1 expression had no effect on glutamatergic inputs to the LA neurons themselves, which supports the idea that circuit development is predominantly modulated by presynaptic/axonal GluK1 KARs at the LA-CeLA projections.

Weakening of glutamatergic transmission at the LA-CeLA projections following GluK1 inactivation was associated with changes in dendritic morphology in the CeLA target neurons, involving reduction in both, the spine density as well as size of the dendritic tree. Likewise, prolonging the developmental KAR activity via GluK1 overexpression resulted in an expansion of the dendritic tree of CeA neurons, consistent with the idea that KAR activity promotes both functional and morphological maturation of the amygdala neuronal networks. The changes in dendritic morphology likely contribute to the observed changes in glutamatergic drive. However, our data does not indicate whether presynaptic KARs directly influence dendritic development of the CeA neurons via some transsynaptic signaling pathway or whether the morphological alterations resulted secondarily from the KAR dependent changes in glutamatergic transmission early in life.

The endogenous KAR activity particularly affected glutamatergic inputs to the subpopulation of late firing neurons in the CeLA, which displayed significantly lower frequency of sEPSCs in preparations where GluK1 activity was knocked out. Since there were no parallel changes in PPR of evoked responses, loss of ongoing KAR dependent regulation of neurotransmitter release is unlikely to explain the functional phenotype in GluK1-deficient amygdala in juvenile/adult stage. Rather, all the available data supports the view that loss of glutamatergic input to the LF neurons is a result of perturbed development of glutamatergic synapses in the absence of GluK1. Excitability of the LF (PKC-δ+) neurons in the CeA controls anxiety –like behaviors in mice (*Tye et al., 2011*; *Cai et al., 2014*) suggesting that impaired early KAR activity, leading to altered excitatory drive to LF CeA neurons, might be a causative factor in developmentally originating anxiety disorders.

In summary, we have here characterized a physiological KAR mechanism regulating formation of the glutamatergic synaptic circuitry in the amygdala. The endogenous KAR signaling operates during the time of intense synaptogenesis, before the onset of amygdala-dependent behaviors and emergence of the cortical inputs. KARs have been implicated in CNS disorders related to dysfunction of the limbic areas, including mood and anxiety disorders, autism, schizophrenia, as well as epilepsy (*Lerma and Marques, 2013*; *Jane et al., 2009*). Indeed, KAR-dependent modulation of the transmission from BLA to CeA is associated with aberrant amygdala-dependent behaviors in adult mice, resembling the human endophenotypes associated with autism and schizophrenia (*Arora et al., 2018*). Our data indicate that the KAR-dependent circuit remodeling physiologically operates in the developing brain, and support the idea that KAR malfunction during circuit development might predispose to neurological disorders later on in life.

## Materials and methods

### Animals

Male and female Wistar rats (P4-P21) were used in most experiments. Heterozygous *Grik1tm1a* (KOMP)Mbp mice in C57BL/6N background were obtained from KOMP repository (UC Davis) and intercrossed to generate *Grik1$^{tm1a/tm1a}$*. *Grik1$^{tm1a/tm1a}$* mice were crossed with CAG-Flp transgenic line to produce a floxed conditional allele (tm1c). Progeny homozygous for the tm1c allele (*Grik1$^{tm1c/tm1c}$*) was used for further breedings and experiments. Loss of *Grik1* mRNA expression after injection of GFP-CRE encoding AAV virus was validated using qPCR and loss of function was confirmed with electrophysiologial analysis (*Figure 6—figure supplement 1*). All experiments were done in accordance with the guidelines given by the ethics committee for animal research at the University of Helsinki.

### RNA extraction, cDNA synthesis and real-time PCR

The BLA was dissected from 500 μm thick sections cut from the brain of Wistar rats as described (*Lauri et al., 2006*). Purification of total RNA, cDNA synthesis and the real-time quantitative PCR was carried out essentially as described (*Orav et al., 2017*) and using the primers listed in *Table 1*. All samples were analyzed in triplicate. The initial mRNA copy numbers of a sample was obtained by relating the Ct of the sample to a standard curve plot. Relative quantification of gene expression at different developmental stages was analysed using standard $2^{-ddCt}$ method.

**Table 1.** Real-time PCR primers

| Target | Forward | Reverse | Size bp |
|---|---|---|---|
| Grik1 | ATGTGACGCAGAGGAACTGC | GCAGTTGAAGAATGGCAATCG | 126 |
| Grik2 | GTTTGTTACACAGCGGAACTG | CAGCTGAAGAATTGCTATGGTG | 127 |
| Grik3 | CATCGATTCCAAGGGCTACG | CGCCACCACTTCTCCTTCAT | 126 |
| Grik4 | GACACCAAGGGCTATGGGAT | ACCACTTCCGCTTCAGAATC | 118 |
| Grik5 | AGTACGGCACTATCCACGCT | CTCCTCTGTGCTCTTGACGA | 128 |
| Neto1 | TCATAGAAGCTGCCCCAAGG | AAGCCAAAGGGTCCATCTCG | 118 |
| Neto2 | TTTGGAAGCTGCTCCTCGTC | TCCAAGTGATCAAACCGGCA | 93 |
| Gadph | CAGTGCCAGCCTCGTCTCATA | TGGTAACCAGGCGTCCGATA | 79 |

## Western blotting

Eight P4 rats and six P14 rats were used for measuring the protein levels of the GluK2/3, and GluK5 subunits in the BLA. BLA was dissected from 500-μm-thick brain sections and the tissue was sonicated in lysis buffer (1% NP-40, 20 mM Tris, pH 8.0, 137 nM NaCl, 10% glycerol, Phosphatase Inhibitor Cocktail; Upstate BIotechnology). Cellular debris was removed by centrifugation. Protein concentrations were measured by a colorimetric DC Protein Assay (Bio-Rad). 60 μg of each sample from each brain region were used in the analysis in triplicate. Proteins were separated on 4–15% Mini-Protean TGX Gel (Bio-Rad) and transferred to PVDF membranes (Millipore). After blocking with 4% non-fat dry milk in PBS for 1 hr, the membranes were incubated overnight at 4°C with specific primary antibodies, diluted in 2% non-fat dry milk in PBS as follows: anti-GluR6/7 (GluK2/3), (1:2000; Millipore), KA2 (GluK5) (1:2000; Millipore), and anti-beta-actin (1:10000; Sigma-Aldrich), which was used as a loading control in all experiments. After washing in PBS-T, membranes were incubated with peroxidase-conjugated goat anti-rabbit (1:5000; Bio-Rad) or goat anti-mouse (1:5000; Bio-Rad) secondary antibodies (2% non-fat dry milk in PBS; 1 hr at RT). After washing in PBS-T, blots were developed using enhanced chemiluminescence with Pierce ECL Western Blotting Substrate (Thermo Scientific). Intensity of the bands was analysed using ImageJ software and normalized relative to the beta-actin.

## In situ hybridization (ISH)

ISH was carried out on 5-μm-thick paraffin sections from rat and mouse (GAD67-GFP knock-in; Tamamaki et al., 2003) brain as described (Wilkinson and Green, 1990). The digoxigenin (DIG)-labeled antisense and sense RNA probes against Grik1, Grik4, Grik5, Neto1 and Neto2 mRNAs, encoding for KAR GluK1, GluK4, GluK5, NETO1 and NETO2, were as previously described (Vesikansa et al., 2012; Orav et al., 2017). Grik2 cDNA was amplified by PCR using the following primers: Forward GGATGTGATCAGTCTCAAGG, Reverse AGCCAGCAGAACATACATCC. The fragment (532 bp) was subcloned into pGEM-T vector (Promega). The plasmid was linearized by restriction digest and used as a template for in vitro transcription with DIG RNA Labeling kit (Roche Diagnostics).

TSA-Plus Cyanine3/Fluorescein System (Perkin Elmer) was used to visualize ISH signal, followed with a standard DAPI staining. In GAD67-GFP knock-in mouse sections, the primary antibody against GFP (rabbit anti-GFP ab290, Abcam) was added after the ISH signal detection to amplify the GFP signal. Stained sections were imaged using Zeiss Axioimager M2 microscope with Axiocam HRc camera.

The specificity of reaction conditions was first tested in P14 rat hippocampal sections, to ensure specific staining with anti-sense probe and to control that there was no unspecific signal with the corresponding sense RNA probes (Vesikansa et al., 2012). The level of mRNA expression in the amygdala nuclei was quantified based on the optical density of the ISH signal within anatomically defined area, which was normalized against the quantity of cells (DAPI staining) using ImageJ software. mRNA expression level in hippocampal CA3 region was examined as a reference. For the histograms, data from rat and mouse sections have been pooled as no major differences in the

expression pattern between the species were detected. At least three sections were analyzed for each animal, and at least three animals were included in each group.

## Electrophysiology

Acute coronal sections (300–400 µm) were prepared from brains of neonatal (P4-P6) or juvenile (P14-P21) rats or GriK1 $^{tm1c/tm1c}$ mice (P21) using standard methods (*Lauri et al., 2006*). After recovery, the slices were transferred to a submerged recording chamber and superfused with extracellular solution (ACSF) containing (mM): 124 NaCl, 3 KCl, 1.25 $NaH_2PO_4$, 1 $MgSO_4$, 26 $NaHCO_3$, 15 D-glucose, 2 $CaCl_2$; 5% $CO_2$/95% $O_2$, at 30°C. Whole-cell patch clamp recordings were performed from BLA and CeA neurons under visual guidance using patch electrodes with resistance of 3–5 MΩ. Uncompensated series resistance (Rs) was monitored by measuring the peak amplitude of the fast whole-cell capacitance current in response to a 5 mV step. Only experiments where Rs <30 MΩ, and with <20% change in Rs during the experiment, were included in analysis.

Kainate and domoate-induced currents were recorded in the presence of antagonists for AMPA, NMDA and GABA-A receptors (2 µM NBQX, 50 µM D-(-)−2-amino-5-phosphonopentanoic acid (D-AP5) and 100 µM picrotoxin (PiTX), respectively) using intracellular solution containing (in mm): 135 K-gluconate, 10 HEPES, 5 EGTA, 4 Mg-ATP, 0.5 Na-GTP, 2 KCl, and 2 $Ca(OH)_2$ (285 mOsm), pH 7.2. The agonists were applied using fast local application to the slice and washed out by fast local application of ASCF; 33 mM KCl was used as a positive control.

Glutamatergic synaptic currents were recorded using Cs-based intracellular solution containing (in mM): 130 $CsMeSO_4$, 10 HEPES, 0.5 EGTA, 4 Mg-ATP, 0.3 Na-GTP, 5 QX-314, 8 NaCl; 280 ± 5 mOsm (pH 7.2). For post-hoc morphological analysis, biocytin (0.4%) was included in the intracellular solution in some of the recordings. Spontaneous miniature EPSCs (mEPSCs) were recorded in the presence of TTX (1 µM), and PiTX (100 µM) at a holding membrane potential of −70 mV. mEPSCs were analyzed with MiniAnalysis 6.0.3 program (Synaptosoft Inc). Events were verified visually, and events with amplitude less than three times the baseline rms noise level were rejected. For time-course plots, detected events were calculated in 120 s bins. Pertussis toxin (PTX) treatment was performed with acute slices at 37°C for 3–5 hr. In these experiments, slices were washed with 1 mL incubation solution containing (in mM): 105 NaCl, 3 KCl, 1 $MgSO_4$, 3.75 $NaH_2PO_4$, 26 $NaHCO_3$, 2 $CaCl_2$, 15 D-glucose, and 25 HEPES (pH = 7.2) with or without 5 µM PTX, and placed into Millicell CM 0.4 µm membrane inserts (Millipore) with 1 mL of the above solution.

Evoked EPSCs were recorded at a holding potential of −70 mV in response to stimulation of LA by a bipolar metal electrode in the presence of PiTX (100 µM) and D-AP5 (50 µM). Stimulus-response curves were obtained by applying stimuli with fixed intensities, between 0.2 and 2.5 mA. Paired pulse responses were evoked with 50 ms inter-pulse interval. For AMPA-NMDA ratio, evoked EPSCs were recorded in the presence of PiTX at −70 and +40 mV. The AMPAR–related component was calculated as the peak of the response at −70 mV, and NMDAR-mediated current was isolated by calculating the average amplitude of the response recorded at +40 mV 50–60 ms after stimulation. Data were collected and analyzed online using LTP software (*Anderson et al., 2012*; www.winltp.com).

Spontaneous glutamatergic and GABAergic synaptic events (sEPSCs and sIPSCs) were recorded using intracellular solution containing (in mM): 135 K-gluconate, 10 HEPES, 2 KCl, 2 $Ca(OH)_2$, 5 EGTA, 4 Mg-ATP, and 0.5 Na-GTP. After obtaining whole cell access, the firing properties of the cell were identified by applying a current injections (600 ms, increments of 20 pA from −100 to 300 pA) in current clamp mode. Then the cell was voltage clamped to −50 mV for recording of spontaneous synaptic activity. AP threshold and delay were analyzed off-line.

For all the electrophysiological data, n number refers to the number of recorded neurons, which in all datasets were collected from at least three (in most cases from at least 6) different animals. The number of animals used in each dataset is indicated in brackets after the n value. Data obtained from the genetically manipulated mice were analyzed by researcher blind to the type of virus infection.

## Morphological analysis

For *post hoc* morphological characterization of biocytin filled neurons, slices were fixed overnight in a 4% paraformaldehyde (4 °C), after which they were washed with 0.01 M phosphate-buffered saline

(PBS) and permeabilized with 0.01 M PBS 0.3% Triton-X 100 (Sigma-Aldrich) for 1.5 hr at room temperature. Streptavidin Conjugate (1:500; A488; Life Technologies) was added to the permeabilization solution and incubated for 4 hr. PBS-washed slices were mounted onto slides and blind-coded for morphological analysis. Dendritic trees and dendritic spines were imaged using a LSM Zeiss 710 confocal microscope (Zeiss Plan Neofluar 20x/0.50 and alpha Plan146 Apochromat 63x/1.46 OilKorr M27 objectives). Dendritic trees were imaged with resolution of 2.3 pixels/μm and Z-stack interval of 1 μm. Spines were imaged with resolution of 15.17 pixels/μm and Z-stack interval of 0.5 μm. Spine density was evaluated on the primary dendrites for P7-11 and on the secondary for P14-21. Actual spine detection was done using the NeuronStudio software to quantify spines in a Z-stack image. Verification of the spine detection was done manually. Values of spine density per cell were used for statistical analysis. Sholl analysis was performed on 2D maximal intensity images acquired from Z-stacks, using the Fiji ImageJ plugin. The number of intersections was estimated with 10 μm bins, and values per cell were used for statistical analysis. Total dendritic length were measured from traced dendrites of each cell.

## Viral transduction in vivo

Lentiviral vectors encoding epitope-tagged GluK1 were as described previously (*Vesikansa et al., 2012*). The shRNA against GluK1 (target sequence: CCTGGACATTATCAGTCTCAA) was subcloned to modified pLKO.1 vector where the puromycin resistance cassette was replaced with GFP under the synapsin-1 promoter (pLKO.1- syn1-EGFP). The lentiviral particles were produced in HEK293T cells and purified as described (*Vesikansa et al., 2012*). The efficacy of the lentiviral shRNA to suppress expression of GluK1 was validated in cultured DRG cells, where GluK1 is endogenously strongly expressed (*Figure 8—figure supplement 1*). Lentiviral particles were injected to the LA area of 4- to 5-day-old rat pups (for shRNA) and 6- to 9-day-old pups (for overexpression) under isoflurane anesthesia. The animals were placed onto stereotaxic frame, the skull was exposed and two or three small holes at each side were done using dental drill. 0.3–0.6 ml of lentiviral suspension was injected into LA region of amygdala. The stereotaxic coordinates for LA were recalculated in the respect to bregma – lambda distance and varied in the following range for P5-6: AP 1.2, 2.0, 2.8 (from bregma), ML 3.8–3.9, DV 3.6–3.8; and for P6-9: AP 2.0, 2.8 (from bregma), ML 4.5–4.6, DV 6–6.2. The wound was stitched and treated with bacibact (Orion Pharma, Finland), sutured and the pup was left to recover with the dam.

AAV serotype eight vectors encoding for GFP (pAAV.CMV.PI.EGFP.WPRE.bGH) and GFP-CRE (pAAV.CMV.HI.eGFP-Cre.WPRE.SV40) were purchased from Addgene (catalog #105530-AAV8 and 105545-AAV8, respectively). The AAV particles were injected to the BLA of neonatal (P2-4) *Grik1*$^{tm1c/tm1c}$ mice under anaesthesia in a stereotaxic frame as described above, except that the injection was done through the skin and skull using the following coordinates: AP 3.8 (from the most rostral point) ML 2.2, DV 2.2.

The specificity of the injection site was visually monitored after preparation of the acute slices for electrophysiological recordings. Only slices with 1) strong GFP signal in the LA and 2) no visible GFP signal in the CeA (lentiviral injections) or estimated CeA GFP intensity <20% of the level in LA (AAV injections) were included in further analysis. The exact infection rate in different amygdala nuclei was quantified using slices from three animals that had passed the visual inspection criteria. The percentage of GFP positive neurons from all DAPI-positive cells in LA, BA and CeA was calculated from Z-stack confocal images (8–12 images with interval of 10 μm).

## Statistical analysis

All statistical analyses were done on raw (not normalized) data using Sigma Plot 11.0 or Graph Pad Prism 8.0.2 software. To analyse differences between groups, Shapiro-Wilk test was used to test for normal distribution and one-way ANOVA with Holm-Sidak post hoc comparison or one-way ANOVA on ranks (Kruskal-Wallis with Dunn's method for pairwise comparison) was then used accordingly. For pharmacological effects, Student's paired two-tailed t-test or Wilcoxon rank sum test was used. The test as well as the corresponding F, H, t and Z values are indicated in the text, with the degree of freedom as subscript. The results were considered significant when p<0.05. All the pooled data are given as mean ± S.E.M.

## Acknowledgements

We thank Prof. Andreas Lüthi and his group members for their expert help with amygdala slice electrophysiology. Kirsi Kolehmainen, Erja Huttu and Outi Kostia are acknowledged for the outstanding technical help. This study was financially supported by the Academy of Finland, Sigrid Juselius Foundation and Jane and Aatos Erkko foundation.

## Additional information

### Funding

| Funder | Grant reference number | Author |
|---|---|---|
| Academy of Finland | SA 297211 | Sari E Lauri |
| Aatos Erkko Foundation | | Sari E Lauri |
| Sigrid Jusélius Foundation | | Sari E Lauri |

The funders had no role in study design, data collection and interpretation, or the decision to submit the work for publication.

### Author contributions

Maria Ryazantseva, Conceptualization, Formal analysis, Investigation, Methodology; Jonas Englund, Johanna Huupponen, Formal analysis, Investigation, Methodology; Alexandra Shintyapina, Investigation, Methodology; Vasilii Shteinikov, Investigation, Contributed significantly to data acquisition during revision of the manuscript; Asla Pitkänen, Supervision, Methodology; Juha M Partanen, Resources, Supervision, Methodology; Sari E Lauri, Conceptualization, Resources, Supervision, Funding acquisition, Project administration

### Author ORCIDs

Maria Ryazantseva (iD) https://orcid.org/0000-0002-8201-817X
Sari E Lauri (iD) https://orcid.org/0000-0002-5895-1357

### Ethics

Animal experimentation: All experiments were done in accordance with the guidelines given by the ethics committee for animal research at the University of Helsinki (license numbers ESAVI/6853/04.10.07/2017 and ESAVI/29384/2019).

### Decision letter and Author response

Decision letter https://doi.org/10.7554/eLife.52798.sa1
Author response https://doi.org/10.7554/eLife.52798.sa2

## Additional files

### Supplementary files

• Transparent reporting form

### Data availability

All data generated or analysed during this study are included in the manuscript and supporting files. Source data files have been provided for all figures.

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
