## [Decision Letter]

**Acceptance summary:**

The manuscript by Ryazantseva et al. demonstrates that presynaptic kainate receptors are important for the correct development of neural circuitry in the amygdala. Specifically, GluK1 subunits acting via metabotropic signalling pathways regulate the formation of inputs from basal and lateral amygdala into central amygdala.

**Decision letter after peer review:**

Thank you for submitting your article "Kainate-type of glutamate receptors regulate wiring of intrinsic glutamatergic connectivity in the amygdala" for consideration by *eLife*. Your article has been reviewed by two peer reviewers, and the evaluation has been overseen by a Reviewing Editor and Laura Colgin as the Senior Editor. The following individual involved in review of your submission has agreed to reveal their identity: Roger L Clem (Reviewer #3).

The reviewers have extensively discussed the reviews with one another and the Reviewing Editor has drafted this decision to help you prepare a revised submission.

Summary:

This manuscript addresses the impact of kainate receptor expression and activity on development of excitatory synapses in rat amygdala projection neurons. The paper investigates mRNA expression, agonist-induced currents, and antagonist effects on synaptic facilitation in projection cell populations in the lateral, basal and centrolateral nuclei during early postnatal development (4-21 days old). Further, they perform viral genetic manipulations of GluK1 expression, targeted to the basal and lateral nuclei, to determine the impact on dendritic morphology and spontaneous synaptic activity of centrolateral spiny neurons. The findings will be of interest to synaptic physiologists and support a model in which developmental refinement of kainate responses shapes amygdalar function. However, there are potential issues with experimental rigor, the findings are rather disparate in terms of how they can be interrelated at the circuit level, and the conclusion that early postnatal kainate receptors in the lateral amygdala are necessary for development of intrinsic (LA→CeA) synaptic connections is not directly tested.

Essential revisions:

1) Much of the description of cell-type expression of GluK receptors in the first part of the Results is anecdotal, and the relevant points about cell specific expression should be either directly shown (figures or tables) or not reported.

2) It is difficult to be confident that Cre expression was restricted to the basolateral amygdala, especially with neonatal injections. Figure 6A seems to contain a lot of punctate signal beyond the BLA, including within the CeA. This weakens the claim that the developmental effects of GluK1 knockout on CeA morphology and electrophysiology can be attributed entirely to presynaptic kainate receptor activity in the BLA. It is critical to the conclusions of the paper that Cre, shRNA and GluK1 overexpression be carefully documented with a series of confocal micrographs and/or cell counts, and criteria for exclusion should be explicitly described. This is of particular concern given the technical challenges with precise stereotaxic injections into deep neonatal brain structures.

3) The title of Figure 6 is "Local inactivation of GluK1 in the BLA in a cKO model perturbs development of glutamatergic projections to CeA", but the figure actually contains no analysis of BLA projections or BLA-specific synaptic responses. Instead, the analysis focuses on spontaneous synaptic transmission, dendritic properties and spine density in CeA neurons. Although these parameters were affected, whether BLA→CeA synaptic connections were specifically involved remains unaddressed.

4) It remains unclear whether some observed effects of GluK1 knockout, knockdown and overexpression could be due to modulation of glutamate release probability rather than changes in synapse density. This issue is compounded in part through some inconsistencies in the authors' presentation. For example, on the second Results page they state "the mean mEPSC frequency, reflecting the density of functional glutamatergic inputs to the recorded neuron…", but then in the very next paragraph they counter their own argument with "..the mean mEPSC frequency might be affected by factors other than synapse number…". Overall, it is not clear whether a presynaptic mechanism is involved or how BLA synaptic connections were specifically affected since directly evoked responses were not examined.

5) Although the results indicate that morphology and synaptic properties of CeA neurons can be bidirectionally regulated by genetic manipulations of GluK1, an unanswered question is whether these effects are specific to this early developmental period and to what extent these effects can be attributed to disruption of a normative developmental process. For example, how does CeA dendritic morphology and spine density normally change during the period when genetic manipulations were performed?

6) Effects of GluK1 genetic manipulations on BLA neurons, where these genetic constructs were expressed, should be examined. Do these manipulations affect dendritic complexity, spine density or synaptic transmission in BLA neurons and, if so, are these effects restricted to transfected cells (i.e. cell autonomous)?

7) Statistical support for results is incomplete. In particular, many effects are reported as% change from early postnatal periods (e.g.% change from P4 levels). However P4 levels were typically quite heterogenous, and the statistical analysis of developmental differences should take this baseline variability into account, and it is not clear that this was done. Overall, it would probably be better to just report raw values, not normalized to a particular time point, given the underlying variability.

[Editors' note: further revisions were suggested prior to acceptance, as described below.]

Thank you for resubmitting your work entitled "Kainate-type of glutamate receptors regulate wiring of intrinsic glutamatergic connectivity in the rodent amygdala" for further consideration by *eLife*. Your revised article has been evaluated by Laura Colgin as the Senior Editor, a Reviewing Editor and two reviewers.

The manuscript has been improved but there are some remaining issues that need to be addressed before acceptance, as outlined below:

The authors have been very responsive to previous concerns, in particular with their inclusion of new analysis and experimental results. The major conclusions now enjoy sufficient support.

Based in part on new results, the authors should revise the text, including the title, to more accurately reflect the experimental results. Throughout the paper and in the title, the results are described as indicating that kainate receptors regulate "intrinsic wiring" of the amygdala. However, the data really do not address very well whether wiring is affected per se, since this would require some analysis of axons, boutons, connection probability, etc. Rather, the data provide evidence only that input/output relation and AMPA:NMDA ratios are affected by kainate receptor expression at the LA→CeA pathway. The best evidence therefore is that kainate receptors regulate postsynaptic efficacy, rather than wiring. Although some morphological changes were described, these were never localized to intrinsic versus extrinsic connections and so their relevance to the LA→CeA pathway remains unclear. The authors should therefore revise their Introduction, Results and Discussion to more accurately reflect what is shown.

---

## [Author Response]

Essential revisions:1) Much of the description of cell-type expression of GluK receptors in the first part of the results is anecdotal, and the relevant points about cell specific expression should be either directly shown (figures or tables) or not reported.

We have modified the text accordingly and removed all statements that are not directly supported by data shown in the Figure 1.

2) It is difficult to be confident that Cre expression was restricted to the basolateral amygdala, especially with neonatal injections. Figure 6A seems to contain a lot of punctate signal beyond the BLA, including within the CeA. This weakens the claim that the developmental effects of GluK1 knockout on CeA morphology and electrophysiology can be attributed entirely to presynaptic kainate receptor activity in the BLA. It is critical to the conclusions of the paper that Cre, shRNA and GluK1 overexpression be carefully documented with a series of confocal micrographs and/or cell counts, and criteria for exclusion should be explicitly described. This is of particular concern given the technical challenges with precise stereotaxic injections into deep neonatal brain structures.

We have now included confocal images and quantified data (cell counts) to document the AAV and lentiviral infection rates in the different amygdala subnuclei (LA, BA and CeA). These data are included in the revised Figures 6 and 8. We also describe the criteria for exclusion in detail in the revised Materials and methods. These data indicate that while some spread of the AAV virus from BLA to CeA was observed, lentiviral infections were highly specific for LA.

3) The title of Figure 6 is "Local inactivation of GluK1 in the BLA in a cKO model perturbs development of glutamatergic projections to CeA", but the figure actually contains no analysis of BLA projections or BLA-specific synaptic responses. Instead, the analysis focuses on spontaneous synaptic transmission, dendritic properties and spine density in CeA neurons. Although these parameters were affected, whether BLA→CeA synaptic connections were specifically involved remains unaddressed.

We have now included new data to test the efficacy of synaptic transmission at the BLA-CeA projections in cKO mice with local inactivation of GluK1 in the BLA. Specifically, we investigated the input/output relationship, paired pulse ratio and AMPA/NMDA ratio of EPSCs, recorded from CeA in response to LA stimulation. We find significant differences in the input-output ratio and AMPA/NMDA ratio of the EPSCs between AAV CRE – injected GluK1 cKO mice and GFP injected controls. However, no difference in the paired pulse ratio was detected. Our data are consistent with weaker LA-CeA excitatory connectivity in cKO mice lacking GluK1 in the BLA. These data are now included in the revised Figure 7.

4) It remains unclear whether some observed effects of GluK1 knockout, knockdown and overexpression could be due to modulation of glutamate release probability rather than changes in synapse density. This issue is compounded in part through some inconsistencies in the authors' presentation. For example, on the second results page they state "the mean mEPSC frequency, reflecting the density of functional glutamatergic inputs to the recorded neuron…", but then in the very next paragraph they counter their own argument with "...the mean mEPSC frequency might be affected by factors other than synapse number…". Overall, it is not clear whether a presynaptic mechanism is involved or how BLA synaptic connections were specifically affected since directly evoked responses were not examined.

We apologize for the confusing statements and have now removed those from the text.

To test whether transmitter release probability or percentage of silent synapses influence the mEPSC frequency in the GluK1 cKO model, we have now examined paired pulse ratio and AMPA/NMDA ratio of evoked responses in AAV CRE vs. GFP injected mice (see also point 3). Our data indicates that in AAV CRE injected GluK1 cKOs, AMPA/NMDA ratio is lower as compared to controls, while there was no significant difference in the PPR. These data suggest that an increase in the proportion of silent synapses might contribute to the loss of mEPSCs in addition to lower density of synapses (revealed by morphological characterization). These data are now included in the revised Figure 7.

5) Although the results indicate that morphology and synaptic properties of CeA neurons can be bidirectionally regulated by genetic manipulations of GluK1, an unanswered question is whether these effects are specific to this early developmental period and to what extent these effects can be attributed to disruption of a normative developmental process. For example, how does CeA dendritic morphology and spine density normally change during the period when genetic manipulations were performed?

We have now included data to show that spine density and dendritic complexity in CeLA neurons significantly increase during development from P6 to P21. Following GluK1 inactivation, the dendritic morphology at P21 remains similar to P6-P10 WT’s (Figure 6—figure supplement 3). In addition, we show that inactivation of GluK1 expression in BLA at a later developmental stage (BLA virus injection P14, ex vivo recording at P35) had no effect on mEPSCs of dendritic morphology of CeLA neurons. These data are now included in the revised Figure 6E-G.

6) Effects of GluK1 genetic manipulations on BLA neurons, where these genetic constructs were expressed, should be examined. Do these manipulations affect dendritic complexity, spine density or synaptic transmission in BLA neurons and, if so, are these effects restricted to transfected cells (i.e. cell autonomous)?

We have now added new data to show that GluK1 deficiency has no significant effect on mEPSCs or dendritic morphology of LA neurons. These data are now included in revised Figure 6B (mEPSCs) and Figure 6—figure supplement 2 (morphology).

7) Statistical support for results is incomplete. In particular, many effects are reported as% change from early postnatal periods (e.g.% change from P4 levels). However P4 levels were typically quite heterogenous, and the statistical analysis of developmental differences should take this baseline variability into account, and it is not clear that this was done. Overall, it would probably be better to just report raw values, not normalized to a particular time point, given the underlying variability.

All the statistical tests are done using raw data, as indicated in the Materials and methods. Also, raw data are shown in the figures. In the text, to make it easier to the reader, data are given as% difference with the variability indicated ( ± sem).

[Editors' note: further revisions were suggested prior to acceptance, as described below.]The manuscript has been improved but there are some remaining issues that need to be addressed before acceptance, as outlined below:The authors have been very responsive to previous concerns, in particular with their inclusion of new analysis and experimental results. The major conclusions now enjoy sufficient support.Based in part on new results, the authors should revise the text, including the title, to more accurately reflect the experimental results. […] The authors should therefore revise their Introduction, Results and Discussion to more accurately reflect what is shown.

We have now revised the title, Introduction, Results and Discussion as suggested. The changes made are indicated in the revised manuscript.